# The genetic landscape of a metabolic interaction

Thuy N. Nguyen [1,2,3,5], Christine Ingle[1,2,3], Samuel Thompson[4,6] &
Kimberly A. Reynolds [1,2,3] ✉

While much prior work has explored the constraints on protein sequence and evolution induced by physical protein-protein interactions, the sequence-level constraints emerging from non-binding functional interactions in metabolism remain unclear. To quantify how variation in the activity of one enzyme constrains the biochemical parameters and sequence of another, we focus on dihydrofolate reductase (DHFR) and thymidylate synthase (TYMS), a pair of enzymes catalyzing consecutive reactions in folate metabolism. We use deep mutational scanning to quantify the growth rate effect of 2696 DHFR single mutations in 3 TYMS backgrounds under conditions selected to emphasize biochemical epistasis. Our data are well-described by a relatively simple enzyme velocity to growth rate model that quantifies how metabolic context tunes enzyme mutational tolerance. Together our results reveal the structural distribution of epistasis in a metabolic enzyme and establish a foundation for the design of multi-enzyme systems.

Enzymes function within biochemical pathways, exchanging substrates and products to generate useful metabolites. This metabolic context places constraints on enzyme velocity—the product of catalytic activity and enzyme abundance. For example, the relative velocities of some enzymes must be coordinated to avoid accumulation of deleterious metabolic intermediates[1–3]. In other instances, optimal enzyme abundance is set by a tradeoff between the cost of protein synthesis and the benefit of efficient nutrient utilization[4–6]. Considered at the pathway scale, metabolic enzymes are often produced in evolutionarily conserved stoichiometric ratios across species[7], providing further indication that relative—not just absolute—enzyme velocity is under selection. More generally, the relationships amongst the velocity of a given enzyme, metabolic flux through a pathway, and cellular growth rate are non-linear and shaped by interactions between pathway enzymes (Fig. 1a). Indeed, a key result of metabolic control theory is that the control coefficient of an enzyme—defined as the fractional change in pathway-level flux given a fractional change in enzyme velocity—depends on the

starting (native) velocity of the enzyme, but *also* on the velocity of all other enzymes in the pathway[8,9]. That is to say, given that enzymes act sequentially to produce metabolites, the effects of mutations on cellular phenotype can be buffered or amplified depending on which enzymatic reactions control metabolic flux. As a consequence, enzyme mutations that are neutral in one context may have profound consequences for metabolic flux and growth rate in the background of variation in another[10–14]. This context-dependence, or epistasis, amongst metabolic enzymes need not be mediated by direct physical binding, but emerges indirectly through shared metabolite pools and a need to maintain flux while avoiding the accumulation of deleterious intermediates[6,11,15].

While prior work has explored how physical protein-protein interactions (binding) constrain protein sequence, the constraints on sequence and enzymatic activity emerging from these sorts of non-binding functional interactions in metabolism remain unclear. How is this biochemically-mediated epistasis organized in the protein structure and reflected in the sequence? A quantitative

[1]The Green Center for Systems Biology, University of Texas Southwestern Medical Center, Dallas, TX 75390, USA. [2]The Lyda Hill Department of Bioinformatics, University of Texas Southwestern Medical Center, Dallas, TX 75390, USA. [3]The Department of Biophysics, University of Texas Southwestern Medical Center, Dallas, TX 75390, USA. [4]Department of Bioengineering and Therapeutic Sciences, University of California, San Francisco, CA 94158, USA. [5]Present address: Form Bio, Dallas, TX 75226, USA. [6]Present address: Department of Bioengineering, Stanford University, Stanford, CA 94305, USA.
✉e-mail: kimberly.reynolds@utsouthwestern.edu

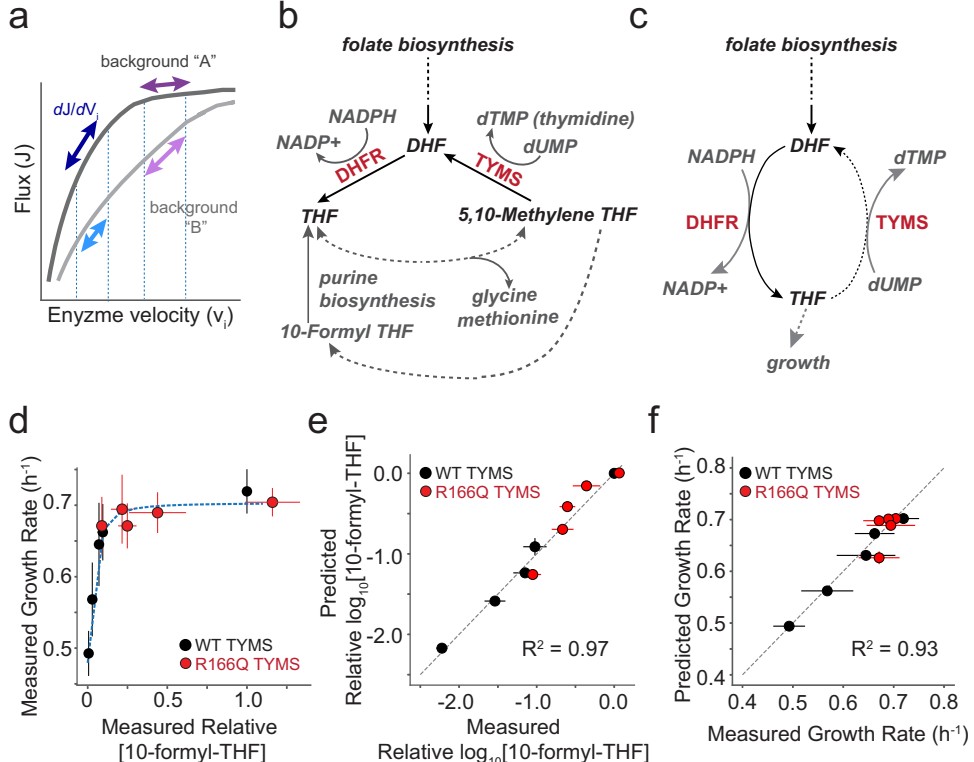

**Fig. 1 | Constructing a biochemistry-to-growth model for DHFR and TYMS.**
**a** Schematic of metabolic control. Many enzymes show a hyperbolic relationship between velocity and flux; the enzyme control coefficient describes the fractional change in flux given a fractional change in velocity. Control coefficients vary with starting enzyme velocity (purple and dark blue arrows, background A) and can change with genetic background (violet and light blue arrows, background B). Consequently mutations can have a strong effect on flux and growth in one background but not another. **b** DHFR and TYMS in folate metabolism. Metabolites are in gray or black italic text. Dotted lines indicate multiple intermediate reactions summarized with a single line. **c** A simplified, abstracted version of the DHFR and TYMS system. Again dotted lines indicate multiple intermediate reactions summarized with a single line. **d** The relationship between experimentally measured relative abundance of [10-formyl-THF] and *E. coli* growth rate. Each point is a particular DHFR/TYMS genotype. Red points indicate five DHFR variants in the background of TYMS R166Q and black indicates the same DHFR variants in the context

of WT TYMS. Error bars indicate the standard deviation across $N = 3$ replicates for growth rate (y-axis) and 10-formyl-THF abundance (x-axis), centered at the mean. The blue dotted line is the best fit hyperbolic model (Eq. (1)) relating THF abundance to growth. **e** Correlation between experimentally measured $\log_{10}$[10-formyl-THF] relative abundance and model prediction (as computed with Eq. (3)). The gray dotted line indicates x = y. Color coding is identical to (**d**). Error bars in x indicate the standard deviation across $N = 3$ replicate experiments (centered at the mean), error bars in y are standard deviation across ten fits obtained by jackknife (leave-one-out) sub-sampling the data and refitting (centered at the mean). **f** Correlation between experimentally measured and predicted growth rates for five DHFR point mutations in two TYMS backgrounds (same mutants as **d**, **e**). The gray dotted line indicates x = y. Color coding is identical to (**d**). Error bars in x indicate the standard deviation across $N = 3$ experimental replicates, error bars in y describe the standard deviation obtained by jackknife (leave-one-out) sub-sampling the data and refitting the model ten times (all points centered at the mean).

understanding of how pathway context shapes sequence and activity would assist in the interpretation of disease-associated mutations, the design of new enzymes, and directing the laboratory evolution of metabolic pathways.

In this work, we examine the residue-level epistatic interactions between a pair of enzymes that catalyze consecutive reactions in folate metabolism: dihdyrofolate reductase (DHFR) and thymidylate synthase (TYMS). The activity of these enzymes is strongly linked to *E. coli* growth rate, they are frequent targets of antibiotics and chemotherapeutics, and they co-evolve as a module both in the laboratory and across thousands of bacterial genomes[1]. Taking this enzyme pair as a simplified model system in which to examine a biochemically-mediated epistatic interaction, we create a mathematical model relating variation in DHFR and TYMS catalytic parameters to growth rate using a focused set of well-characterized point mutants. Then, to more deeply test this model and comprehensively map the pattern of epistasis between these two enzymes, we measure the effect of nearly all possible DHFR single mutations (2696 in total) in the context of three TYMS variants selected to span a range of catalytic activities. The model predicts—and the data

shows—that TYMS background profoundly changes both the sign (buffering vs. aggravating) and magnitude of DHFR epistasis. Mapping the epistatic effects of mutation to the DHFR tertiary structure reveals that they are organized into distinct clusters based on epistatic sign. Additionally, mutations with the largest magnitude epistatic effect to TYMS center around the DHFR active site, while more weakly epistatic positions radiate outwards. Finally, we infer approximate values for DHFR catalytic power ($k_{cat}/K_m$) across all 2696 mutations by using growth rate measurements across TYMS backgrounds to constrain the enzyme velocity to growth rate model. The residues linked to catalysis form a structurally distributed network inside the enzyme and are highly evolutionarily conserved. Taken together, our data demonstrates at single-residue resolution how epistasis mediated through a biochemical interaction reshapes a mutational landscape. Our results indicate that metabolic context can strongly modulate enzyme evolution in both the clinic and the lab by facilitating or frustrating available mutational paths. More generally, our results invite one to consider new ideas for the joint design of multi-enzyme systems that take into account shared constraints on relative activity and sequence.

## Results

### An enzyme velocity to growth rate model for DHFR and TYMS

We constructed a mathematical model relating changes in DHFR and TYMS catalytic parameters to growth rate phenotype. Our goals were to (1) formalize our previous empirical findings describing DHFR/TYMS biochemical coupling[1], (2) quantify the absolute and relative constraints on DHFR and TYMS catalytic activity, and (3) define the relationship between biochemical activity and epistasis. DHFR and TYMS play a central role in folate metabolism, a well-conserved biochemical pathway involved in the synthesis of purine nucleotides, thymidine, glycine, and methionine[16] (Fig. 1b). Consequently, this pathway is strongly linked to cell growth and a frequent target of antibiotics and chemotherapeutics. DHFR is a 159 amino acid enzyme that catalyzes the reduction of dihydrofolate (DHF) to tetrahydrofolate (THF) using NADPH as a cofactor. The reduced THF then serves as a carrier for activated one-carbon units in downstream metabolic processes. TYMS catalyzes the oxidation of THF back to DHF during deoxythymidine synthesis and is the sole enzyme responsible for recycling the DHF pool[17,18]. Prior work by ourselves and others indicates that these two enzymes are strongly functionally coupled to each other and less coupled to the remainder of the pathway: they co-evolve with each other in terms of synteny and gene co-occurrence across bacterial species (while far less so with the remainder of the pathway)[1], suppressor mutations in TYMS are sufficient to rescue inhibition of DHFR with trimethoprim in both the lab and the clinic[1,19], and reduced expression of DHFR is uniquely rescued by reduced expression of TYMS (and no other folate metabolic enzyme)[1,20,21]. Metabolomics data indicated that loss of DHFR function resulted in accumulation of DHF and depletion of reduced folates, and that compensatory loss of function mutations in TYMS help to restore DHF and THF pools to more native-like levels[1,22,23]. Thus, DHFR and TYMS are a growth-linked two-enzyme system where epistasis is driven by a biochemical interaction, with the added simplification that they are relatively functionally decoupled from surrounding metabolic context.

With this information in mind we defined a mathematical model comprised of two parts: the relationship between intracellular THF abundance and growth rate, and the relationship between enzymatic activity and intracellular THF. First, we considered the relationship between intracellular THF abundance and growth rate. THF limitation due to DHFR loss of function restricts the production of several growth-linked factors, including thymidine, methionine, glycine, and the purine precursors inosine and AICAR. Under the experimental conditions of our growth rate assays—M9 minimal media with 0.4% glucose, 0.2% amicase, and 50 μg/ml thymidine—thymidine is not growth limiting (TYMS R166Q is rescued to WT-like growth) and amicase provides a source of free amino acids. These conditions—which remove selection pressure on TYMS due to thymidine production—emphasize coupling between DHFR and TYMS through the shared THF pool which must be used to produce purine nucleotides. We previously observed a hyperbolic dependence of growth rate on reduced folate abundance for many THF species in these experimental conditions[1]. We selected 10-formyl THF with three glutamates as a representative growth-linked reduced folate for parameterizing the model given it's clear relationship to growth and proximity to purine biosynthesis. Following a similar approach as Rodrigues et al, we used a single four-parameter sigmoidal function to relate growth rate to the experimental measurements of intracellular THF concentration[24].

$$g = \frac{g_{max} - g_{min}}{1 + (K/[\text{THF}])^n} + g_{min} \qquad (1)$$

Here, $g_{max}$ represents the maximal growth rate, $g_{min}$ is the minimal growth rate, $K$ is a constant that captures the concentration of THF that yields 50% growth, and $n$ is a Hill coefficient (Supplementary Table 1).

The second piece of the model connects variation in DHFR and TYMS enzyme velocity to intracellular THF concentrations. To simplify our model, we reduced the pathway to a cycle in which DHFR and TYMS catalyze opposing oxidation and reduction reactions (Fig. 1c). This abstraction assumes that DHFR and TYMS dominate turnover of the DHF and THF pools, and that the reduced folates are considered as a single THF pool. We omitted downstream reactions of folate metabolism that use one-carbon derivatives of THF in the production of purine precursors, glycine, and methionine (dashed lines in Fig. 1b), as none of these other reactions oxidize THF back to DHF—they instead add or subtract one-carbon units from the reduced THF. We thus treated the intracellular concentration of total folate ([DHF] + [THF]) as a constant, with DHFR and TYMS activity setting the balance between the reduced and oxidized forms. This simplification formalizes the notion that DHFR and TYMS are a two-enzyme module tightly coupled to each other but less so to the remainder of the pathway as indicated by our prior comparative genomics and laboratory evolution experiments[1]. Given this abstraction, we write a rate equation that isolates the recycling of THF in terms of a small number of measurable biochemical parameters:

$$\frac{d[\text{THF}]}{dt} = \frac{[\text{DHFR}]^* k_{cat}^{DH}}{1 + K_m^{DH}/([\text{fol}_{tot}] - [\text{THF}])} - \frac{[\text{TYMS}]^* k_{cat}^{TS}}{1 + K_m^{TS}/([\text{THF}])} \qquad (2)$$

In this equation, DHFR and TYMS are treated as catalyzing opposing reactions with Michaelis Menten kinetics, providing a relationship between steady state kinetics parameters ($k_{cat}^{DH}$, $K_m^{DH}$, $k_{cat}^{TS}$, $K_m^{TS}$) and intracellular THF abundance. From this equation one can find an analytical solution for the steady state concentration of THF in the form of the Goldbeter-Koshland equation[25,26].

$$\frac{[\text{THF}_{ss}]}{[\text{fol}_{tot}]}$$

$$= \frac{\frac{V_1}{V_2}*\left(1 - \hat{K}_{m2}\right) - \hat{K}_{m1} - 1 + \sqrt{4\hat{K}_{m2}\frac{V_1}{V_2}\left(\frac{V_1}{V_2} - 1\right) + \left(\frac{V_1}{V_2}\left(\hat{K}_{m2} - 1\right) + \hat{K}_{m1} + 1\right)^2}}{2*\left(\frac{V_1}{V_2} - 1\right)}$$

$$(3)$$

Where:

$$V_1 = [\text{DHFR}]k_{cat}^{DH} \quad \hat{K}_{m1} = K_m^{DH}/\text{fol}_{tot} \quad V_2 = [\text{TYMS}]k_{cat}^{TS} \quad \hat{K}_{m2} = K_m^{TS}/\text{fol}_{tot}$$

$$(4)$$

To parameterize the complete model, we used a previously collected set of metabolomics and growth rate data for five DHFR point mutants in the background of both WT TYMS and TYMS R166Q (Supplementary Table 2, Supplementary Table 3)[1]. The five DHFR point mutations were selected to span a wide range of catalytic activities. TYMS R166Q is an active site mutation with near complete loss of catalytic activity[27]. The steady state catalytic parameters ($k_{cat}^{DH}$, $K_m^{DH}$, $k_{cat}^{TS}$, $K_m^{TS}$) were experimentally measured in vitro using purified samples of all DHFR and TYMS variants, with the exception of TYMS R166Q which is near-inactive and assigned an arbitrarily low $k_{cat}$ and high $K_m$ (Supplementary Table 2, Supplementary Table 3). Additionally, the relative abundance of 10-formyl-THF was previously measured by liquid chromatography mass spectrometry for all ten DHFR/TYMS combinations[1]. Given these data, we first fit the relationship between relative 10-formyl-THF abundance and growth rate for these 10 genotypes (Eq. 1), yielding four parameter values for $g_{max}$, $g_{min}$, $K$, and $n$ (Fig. 1d, Supplementary Table 1). Four more fit parameters then remain in Eq. 3: (1) the concentration of the total folate pool ([fol$_{tot}$]) (2) the intracellular concentration of DHFR ([DHFR]), (3) the intracellular concentration of WT TYMS ([TYMS$_{WT}$]), and (4) the intracellular concentration of TYMS R166Q ([TYMS$_{R166Q}$]). We chose to fit the concentration of each TYMS variant separately, while defining a single

global parameter for the concentration of all DHFR variants (which was held constant regardless of variant identity). This means that our model does not capture the differential impact of mutations on DHFR intracellular abundance. Our logic was that TYMS variant concentration should be better constrained by our data, because our experiments quantify the effect of relatively few TYMS variants in many DHFR genetic backgrounds. In contrast, DHFR variant concentration should be less well constrained as our dataset includes the effect of each DHFR variant in only a few TYMS genetic backgrounds. This relatively simplified model showed good correspondence to the data when fit ($R^2 = 0.96$, Fig. 1e, Supplementary Table 1). Equations 1 and 3 were combined to estimate growth rate as a function of both DHFR and TYMS activity, by linking catalytic activity to THF abundance, and then THF abundance to growth rate. The complete model worked well to predict growth rate on our initial training set (Fig. 1f).

## TYMS context alters the sign and magnitude of DHFR mutational effects

To more rigorously test our model and understand its predictions, we expanded our dataset to include more DHFR and TYMS variants with experimentally characterized activities. As our initial model was developed using only two extreme TYMS variants (wild-type and a near complete loss of function variant, R166Q), we were particularly curious to evaluate model performance for TYMS mutations with intermediate effects on catalysis and *E. coli* growth. We identified candidate TYMS mutations by examining an earlier growth complementation study[28]. A handful of these mutants were then cloned, screened for expression, and when possible, purified and characterized. Through this miniscreen we selected two mutations that stably expressed, purified robustly, and yielded intermediate activities: TYMS R127A and Q33S (Fig. 2a). The R127A mutation is located in the TYMS active site and is one of four arginines that coordinate the substrate (dUMP) phosphate group. The Q33S mutation is located at the TYMS dimer interface, distal to the active site. We observed that R127A was more deleterious to catalytic function than Q33S, but that both mutations were more active than R166Q (which shows almost no measurable activity in vitro, Fig. 2b, Supplementary Table 3).

We measured growth rates for seven catalytically characterized DHFR variants (a set of single and double mutants selected to span a range of catalytic activities) in the background of these four TYMS mutants (WT, R127A, Q33S and R166Q). The point mutants were created in a plasmid encoding both DHFR and TYMS (see Methods for details), and the plasmids were transformed into an E. coli selection strain lacking the endogenous DHFR and TYMS genes (ER2566 *ΔfolA ΔthyA*). Growth rates were measured in triplicate using a plate-reader-based assay (28 measurements total; Fig. 2c, Supplementary Fig. 1a,c). We used this focused dataset to re-parameterize the model equations, this time fitting five total parameters ([fol$_{tot}$],[DHFR],[TYMS$_{WT}$],[TYMS$_{Q33S}$],[TYMS$_{R127A}$],[TYMS$_{R166Q}$], Supplementary Table 1). This second round of fitting tested the ability of growth rate data alone to constrain the model—an important step because metabolomics data are available for only a limited number of DHFR and TYMS mutants and are inherently far lower throughput to collect than growth rates. This iteration of parameterization also tested the capacity of the model to capture TYMS mutations with intermediate effects on activity. The data were again well described by the model (Fig. 2C, Supplementary Fig. 1b–d). The best fit parameters result in a growth rate plateau near 0.9 for the most fit mutant combinations (experimental growth rates between 0.7 and 1.0), but allow resolution of growth rates amongst the least fit mutant combinations (growth rates less that 0.7). We observed some variation in the fit parameters (relative to the older data in Fig. 1); this difference might be attributed to the fact that our newer experiments used a revised selection vector backbone.

As a control for overfitting, we tested the ability of the model to predict growth rates for arbitrary catalytic data. We randomly shuffled the catalytic parameters ($k_{cat}$ and K$_m$) among mutations for both DHFR and TYMS, refit all free model parameters, and calculated the RMSD and $R^2$ values between the best fit model and the shuffled data. Importantly, the model was generally unable to describe the experimental growth rate data when catalytic parameters were shuffled across both DHFR and TYMS (Supplementary Fig. 1e, f). This indicated that the model provided a specific description of our experiment and was not trivially overfit. The model was less sensitive to shuffling TYMS catalytic parameters (presumably because we included fit parameters describing the abundance of each TYMS mutation that can compensate for this shuffling, Supplementary Fig. 1h). However, it was strongly sensitive to shuffling DHFR parameters (Supplementary Fig. 1g). Taken together, this analysis indicated that the model provides a good description of the enzyme-velocity-to-growth-rate relationship and can be used to predict and interpret how molecular changes in DHFR and TYMS activity modulate growth rate phenotype.

To examine the model more closely, we considered the relationship between TYMS catalytic activity and DHFR mutational sensitivity. As in previous work, we observed that loss-of-function mutations in DHFR can be partly or even entirely rescued by the loss-of-function mutation TYMS R166Q in the presence of thymidine (Supplementary Fig. 1a–c). TYMS R127A, a less severe loss of function mutation, showed a similar albeit more modest trend—this mutation was able to partly rescue growth for some (though not all) DHFR mutations. A central factor behind DHFR and TYMS biochemical coupling is that loss-of-function mutations in TYMS help to preserve reduced folate pools, allowing THF to shuttle one-carbon units in downstream biochemical processes like purine biosynthesis even when DHFR activity is low. Moreover, loss of TYMS activity reduces accumulation of DHF, which can inhibit upstream reactions[1,23]. Thus, the TYMS R166Q and R127A variants show positive (buffering) epistasis to low-activity DHFR mutations.

In contrast to our expectation that a more intermediate mutation would also demonstrate intermediate levels of positive (buffering) epistasis, TYMS Q33S showed negative epistasis to some DHFR mutations (Supplementary Fig. 1c). This means that these DHFR mutations are more deleterious in the background of TYMS Q33S than in the native TYMS context. To account for this observation, the model predicted that the intracellular concentration of TYMS Q33S is increased relative to wildtype by about three fold (a fit parameter, Supplementary Table 1) such that the V$_{max}$ of TYMS Q33S becomes greater than wildtype ([TYMS$_{Q33S}$]$k_{cat}^{TYMS-Q}$33S>[TYMS$_{WT}$]$k_{cat}^{TYMS-WT}$). This in turn increased the intracellular requirement for DHFR activity, resulting in negative epistasis. To test the prediction that TYMS Q33S is more highly expressed and thus attains a higher intracellular velocity even though less catalytically active, we conducted a series of enzyme kinetics assays using cell lysates[24]. In these experiments we monitored the accumulation of DHF spectrophotometrically in cell lysates (extracted from early log phase cells grown in conditions identical to our selection) following addition of saturating amounts of the TYMS substrates deoxyuridine monophosphate (dUMP) and N5,N10-methylene tetrahydrofolate (MTHF) (see also Methods). DHF accumulation over time was compared for WT ER2566 lysates, the ER2566 DHFR/TYMS double knockout strain (ER2566 *ΔfolA ΔthyA*), and strains carrying selection plasmids encoding TYMS Q33S, TYMS R127A, and TYMS R166Q (Fig. 2d). Importantly, as our plasmid system encodes both DHFR and TYMS, we mutated the DHFR of these plasmids to the near-catalytically-inactive variant D27N. This prevents DHF from being recycled back to THF by DHFR (which would confound our estimations of intracellular TYMS concentration). In these experiments, we observed that all plasmid-containing strains exhibited TYMS activity on par with or less than the native ER2566 strain, indicating that our

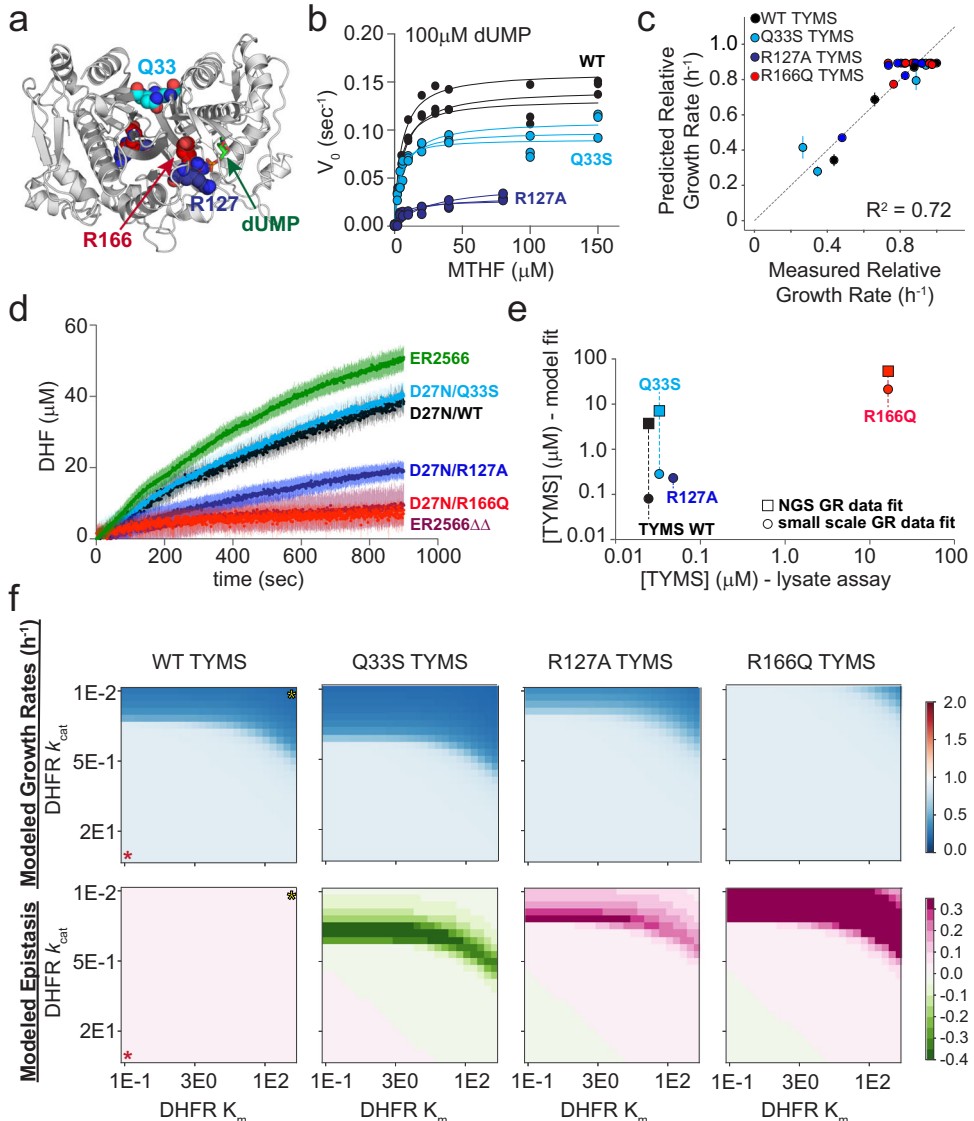

**Fig. 2 | TYMS catalytic activity modulates the epistatic constraints on DHFR sequence and catalytic activity. a** Location of the TYMS point mutations (PDBID: 1BID[60]). TYMS is an obligate domain-swapped homodimer; active sites include residues from both monomers (white and gray cartoon). Positions mutated in this study are in colored spheres (Q33−cyan, R127−navy, R166−red). The dUMP substrate is in green sticks. **b** Michaelis Menten enzyme kinetics for WT TYMS (black), TYMS Q33S (cyan), and TYMS R127A (navy). Experimental replicates (3 total) are plotted individually. Points indicate experimental data and lines the best fit steady state model. **c** Correlation between experimentally measured and model-predicted relative growth rates for seven DHFR variants in four TYMS backgrounds. Each point represents one DHFR/TYMS genotype. Error bars in the x direction are SEM across triplicate growth rate measurements, error bars in y are the SEM estimated from jackknife (leave-one-out) sub-sampling the data and refitting the model. The points themselves are mean-centered. **d** TYMS lysate assays. Curves indicate DHF accumulation over time following substrate addition to crude lysates of WT ER2566

(green), ER2566 ΔfolA ΔthyA (maroon), or ER2566 ΔfolA ΔthyA cells containing plasmid-encoded DHFR D27N paired with TYMS Q33S (cyan), TYMS WT (black), TYMS R127A (navy), or TYMS R166Q (red). Each curve represents the mean of three replicates, the shaded region describes standard deviation. **e** Correlation between intracellular TYMS concentrations determined by model fit and lysate assay. TYMS color coding follows from (**a**–**d**). Circles show model fit values obtained from the small scale plate reader assay (as in (**c**)), while squares show fit values obtained from the deep mutational scan (as in Fig. 5a–c). TYMS R127A was not included in the deep mutational scan. **f** Heatmaps of simulated growth rates (top row) and epistasis (bottom row) computed over a range of DHFR kinetic parameters in four TYMS backgrounds. In the left-most column a red star marks the highest activity enzyme (low $K_m$, high $k_{cat}$), while a yellow star marks the lowest activity enzyme. Growth rates are indicated with a blue-white-red color map, where a relative growth rate of one (white) is equivalent to WT. Epistasis values are indicated with a green-white-pink color map, where zero epistasis is shown in white.

TYMS enzyme is not strongly overexpressed. Moreover, these data showed that the TYMS activity of Q33S-containing lysates is indeed modestly higher than WT TYMS-containing lysates as predicted by our model. By dividing the measured lysate velocity (in Fig. 2d), by the $k_{cat}$ (from Fig. 2b), we obtain estimates of intracellular TYMS concentration and find that these are well-correlated to our model predictions (Fig. 2e).

To further explore the pattern of epistasis across TYMS backgrounds, we simulated growth rates over a range of DHFR $k_{cat}$ and $K_m$

values in each TYMS background (Fig. 2f). This provided a comprehensive prediction of the TYMS-induced constraints on DHFR activity. In particular, we obtained a regime of DHFR $k_{cat}$ and $K_m$ values that is sufficient to support growth for each TYMS mutation. From these data we computed epistasis as the difference in growth rates between a given TYMS background and the WT (see also methods). We observed that TYMS Q33S has negative epistasis to DHFR variants spanning a well-defined band of catalytic parameters. This ridge of negative epistasis (green band in Fig. 2f) describes a range of catalytic

parameters that are sufficient to rescue growth in the context of WT TYMS, but which are deleterious in the context of TYMS Q33S. DHFR variants below this band (with increased $k_{cat}$) grow well in both contexts, while DHFR variants above this band (with decreased $k_{cat}$) grow poorly in both contexts. The simulations also indicated that R127A has weak positive epistasis over a regime of moderately impaired DHFR variants, but R127A is insufficient to rescue growth for the strongest loss of function variants. Finally, TYMS R166Q was observed to be broadly rescuing; DHFR variants need only a negligible amount of activity to support growth in this context. Together, our simulations showed that the sign and magnitude of DHFR epistasis are strongly tuned by TYMS background, and provided quantitative predictions of the catalytic regimes where epistasis is most apparent.

## The single-mutant landscape of DHFR is strongly modulated by TYMS

Next we examined the structural pattern of biochemical epistasis at the residue level across DHFR. This also provided an opportunity to see if the model predictions—negative epistasis for Q33S and broadly positive epistasis for R166Q—held true across a larger dataset. To accomplish this, we created a plasmid-based saturation mutagenesis library of DHFR containing all possible single mutations at every position (3002 total). This library was subcloned into all three TYMS backgrounds. Sequencing showed that these libraries are well-distributed and approach full coverage of all single mutations (97.1%—WT TYMS, 94.6%—TYMS Q33S, 99.3%—TYMS R166Q) (Supplementary Fig. 2). We transformed these libraries into the *E. coli* selection strain (ER2566 *ΔfolA ΔthyA*). Transformants for each library were then grown as a mixed population in selective media in a turbidostat to ensure maintenance of exponential growth and constancy of media conditions. By quantifying the change in the relative frequency of individual mutants over time with next generation sequencing, we obtained a growth rate difference relative to WT DHFR for nearly all mutations in the library (Fig. 3a, Supplementary Data 1, see methods for details). All relative growth rate measurements were made in triplicate, with good concordance among replicates (Supplementary Fig. 3).

The entire dataset showed that the DHFR mutational landscape was strongly dependent on TYMS background (Fig. 3b–e). In all three TYMS backgrounds, the distribution of growth rate effects was bimodal and reasonably well-described by a double gaussian containing one peak of near-neutral mutations and another (far smaller) peak of mutations with highly deleterious growth rate effects. This is the expected result for an enzyme that shows a sigmoidal relationship between activity and growth. In the native TYMS context, the vast majority of mutations fall into the near-neutral peak. However, there is a substantial fraction (12%, 343 total) that display growth rates at or below that of inactive, where inactive was defined as the average growth rate across nonsense mutations in the first 120 residues of DHFR. Consistent with expectation, mutations at known positions of functional importance tended to be deleterious in the WT TYMS context (W22, D27, F31, T35, L54, R57, T113, G121, and D122)[29]. For example, both W22 and D27 are directly in the active site and serve to coordinate substrate through a hydrogen bonding network[30], G121 and D122 are part of the βf-βg loop and stabilize conformational changes associated to catalysis[31,32], and F31 contacts the substrate and is associated to the network of promoting motions[33,34]. In the TYMS Q33S context, many of these deleterious mutations had even more severe effects or were classified as Null. Null mutations disappeared from our sequencing counts within the first three time points (8 h) of the selection experiment, preventing accurate inference of growth rate. For example, mutations at position 22 are deleterious in the WT TYMS context, and appear as Null or very deleterious in the Q33S context. The same pattern can be readily observed for positions 7,14,15, 22, 27, 31, 35, and 121. In contrast, multiple observations are consistent with TYMS R166Q broadly buffering DHFR variation. First, in the TYMS

R166Q context, there are very few deleterious mutations. Nearly all mutations are contained in the near-neutral peak, including mutations at highly conserved active site positions like M20, W22, and L28. Stop codons and mutations at the active site residue D27 continued to be deleterious, indicating that DHFR activity was still under (very weak) selection in the TYMS R166Q background. Second, we observed an average of 41 null mutations per experimental replicate in the WT TYMS context, 82 null mutations in the TYMS Q33S context, but only 7 null mutations per experimental replicate in the R166Q TYMS context. Third, for the TYMS Q33S context as with WT TYMS, we saw that 12% of mutations have growth rates at or below that of inactive variants while only 5% of mutations displayed growth rates at or below those of inactive mutations in the TYMS R166Q context.

To quantify the context dependence of mutational effects, we computed epistasis relative to WT TYMS for all DHFR mutations with measurable relative growth rates in each of the three TYMS backgrounds (2696 in total, see also methods) (Fig. 4, Supplementary Fig. 4, Supplementary Data 2). We assessed the statistical significance of epistasis by unequal variance t-test under the null hypothesis that the mutations have equal mean growth rates in both TYMS backgrounds. These $p$ values were compared to a multiple-hypothesis testing adjusted p-value determined by Sequential Goodness of Fit ($P = 0.035$ for TYMS Q33S and $P = 0.029$ for TYMS R166Q, Fig. 4a, b)[35]. In the TYMS Q33S background, 95 mutations (3%) showed significant negative epistasis and 280 mutations (9%) showed significant positive epistasis. Many of the DHFR mutations with positive epistasis to Q33S were near-neutral in the WT context, and displayed small improvements in growth rate that were highly significant due to the low experimental error for the best-growing mutations (Fig. 4c). In contrast, the mutations with negative epistasis exhibited a range of growth rate effects in the WT context. For the TYMS R166Q background the overall proportion of significant epistatic mutations was larger: while only 41 mutations (1%) showed significant negative epistasis, 851 mutations (28%) showed significant positive epistasis. A smaller number of deleterious mutations in the WT context were not rescued by R166Q (gray points with relative growth rates near 0.25). One possibility is that these mutations are deleterious for reasons beyond disrupting metabolic flux; for example they may result in protein aggregation or off-target physical interactions. Regardless, direct comparison of the relative growth rates of mutations across the WT, Q33S, and R166Q TYMS backgrounds makes it very obvious that TYMS R166Q was broadly rescuing, while TYMS Q33S had a more subtle effect that sometimes yielded negative epistasis (Fig. 4c, d).

## The velocity-to-growth model captures observed fitness landscapes

Next we sought to further test our enzyme velocity to growth-rate model using the deep mutational scanning data. We refit the model a third time, in order to include all available experimental information in parameterizing the model for this larger set of mutants. This included a larger dataset of 34 DHFR single mutants with previously reported $k_{cat}$ and $K_m$ values. We additionally characterized $k_{cat}$ and $K_m$ for four new DHFR mutations (I5K, V13H, E17V and M20Q) that exhibited strong sign epistasis to TYMS to more completely test our ability to predict epistasis. Together this yielded a set of 114 growth rate measurements with matched $k_{cat}$ and $K_m$ values for DHFR and TYMS (38 DHFR mutations in 3 TYMS backgrounds, Supplementary Table 2). We used these data to perform a bootstrap analysis; iteratively subsampling the data and refitting the model 1000 times to obtain standard deviations in our model fit and the eight associated parameters (Fig. 5a). The inferred parameters for this large set of sequencing-based growth rate measurements were qualitatively similar to those obtained for the smaller set of 28 plate-reader based growth rate measurements (7 DHFR mutants in 4 TYMS backgrounds, Fig. 2), but we observed some discrepancy in the estimated total folate pool and intracellular

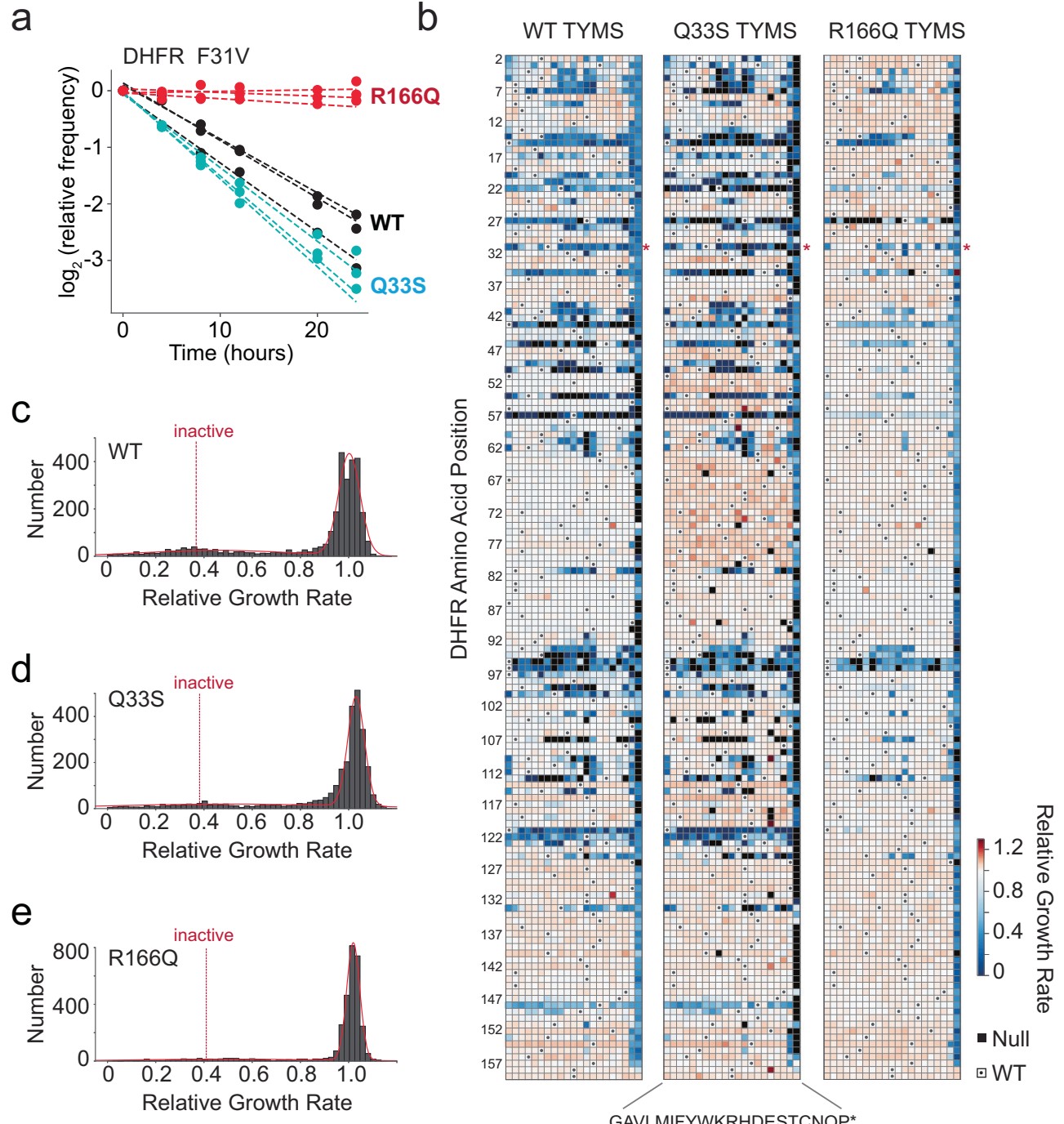

**Fig. 3 | The effects of DHFR mutation on growth rate in three TYMS backgrounds. a** Sequencing-based growth rate measurements for DHFR F31V in three TYMS backgrounds: R166Q (red), Q33S (cyan), and WT (black). Each point represents one triplicate experimental measurement. Dotted lines indicate linear regression fits to each replicate, the slope of each line is the inferred growth rate (relative to WT) for that DHFR/TYMS mutant combination. **b** Heatmaps of the growth rate effect for all DHFR single mutations. DHFR positions are along the horizontal axis; amino acid residues (along the vertical axis) are organized by physiochemical similarity. The displayed relative growth rate is an average across three replicates, and is normalized such that the WT DHFR is equal to one. Red indicates mutations that increase growth rate, white indicates mutations with wild-type like growth, and blue indicates mutations that decrease growth rate. Null mutations (black squares) were not observed by sequencing after the first two time points, and thus there was insufficient data for growth rate inference. Small dots mark the WT residue identity in each column. **c** The distribution of DHFR mutational effects in the WT TYMS background. The red line indicates a best-fit double gaussian, gray bars are the data. The red, dashed inactive line marks the average relative growth rate for nonsense mutations (stop codons) in the first 120 positions of DHFR. The WT DHFR growth rate is equal to one. **d** The distribution of DHFR mutational effects in the TYMS Q33S background, color coding identical to (**c**). **e** The distribution of DHFR mutational effects in the TYMS R166Q background, color coding identical to (**c**). Note that the y-axis for (**e**) is distinct from (**c**) and (**d**).

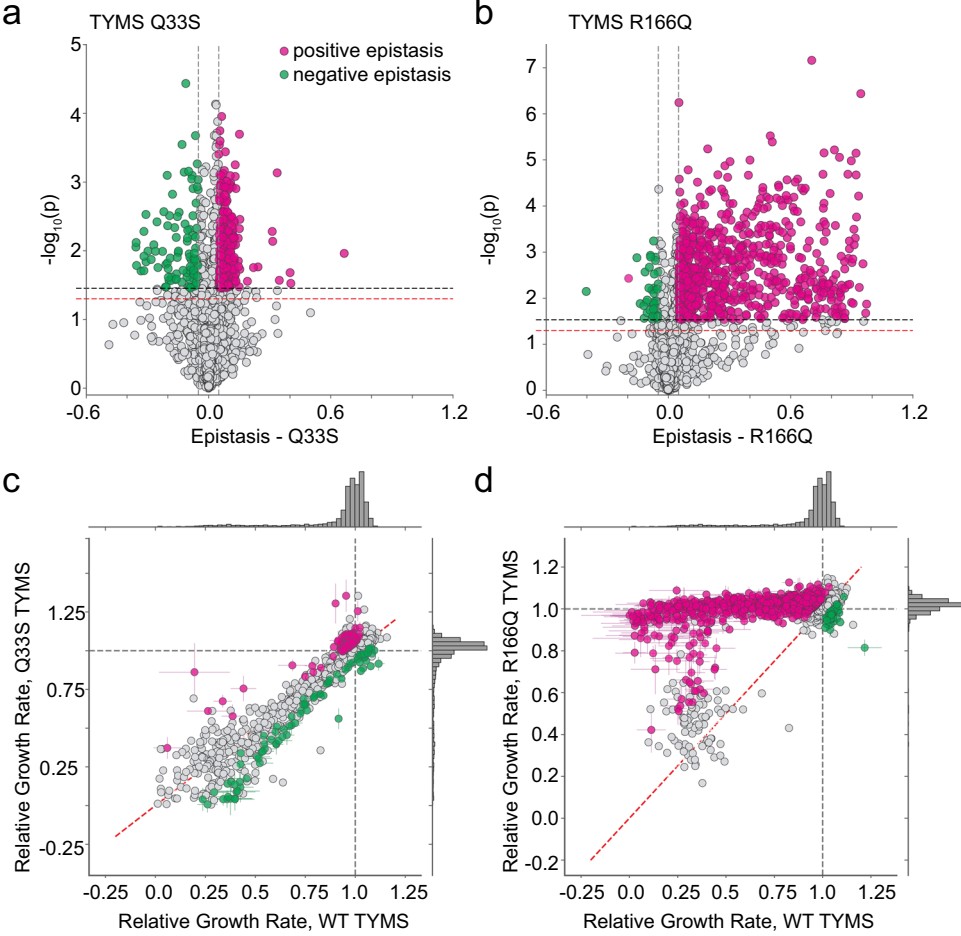

**Fig. 4 | Epistatic coupling of DHFR to two TYMS backgrounds. a** Volcano plot examining the statistical significance of epistasis across all DHFR point mutations in the Q33S background. $P$ values were calculated by unequal variance two-sided $t$ test under the null hypothesis that the mutations have equal mean growth rates in both TYMS backgrounds (across triplicate measurements). The red horizontal dashed line marks the standard significance cutoff of $P = 0.05$, the black horizontal dashed line indicates a multiple-hypothesis testing adjusted $p$ value ($P = 0.035$). The gray vertical dashed lines indicate an empirical threshold for epistasis. Pink and green indicate statistically significant positive and negative epistasis respectively.
**b** Volcano plot examining the statistical significance of epistasis across all DHFR point mutations in the R166Q background. $P$ values were calculated as in (**a**); the multiple-hypothesis testing adjusted p-value for the R166Q background was

($P = 0.029$). Color coding is identical to (**a**). **c** Comparison of the relative growth rate effects for DHFR single mutants in the WT and TYMS Q33S backgrounds. The marginal distribution of growth rate effects is shown along each axis. Mutations with statistically significant positive and negative epistasis are indicated in pink and green respectively. The WT relative growth rate equals one, and is indicated with a dashed gray line across each axis. The dashed red line marks x = y. Error bars (in x and y) indicate standard deviation across three experimental measurements, points are centered at the mean. **d** Comparison of the relative growth rate effects for DHFR single mutants in the WT and TYMS R166Q backgrounds. Plot layout and color coding is identical to (**c**). Error bars (in x and y) indicate standard deviation across three experimental measurements, points are centered at the mean.

concentrations of TYMS (Supplementary Table 1). In general, the total folate pool and concentration of TYMS R166Q were more variable across the bootstrap analysis, indicating that these two parameters are less well constrained by our data (as indicated by the estimated variances in Supplementary Table 1). Nevertheless, the inferred parameters in both the plate reader and sequencing-based experiments suggested similar relative expression levels for DHFR and the three TYMS point mutants, with Q33S being roughly 2–3 times more abundant than WT TYMS. These model-fit concentrations of TYMS correlated with the estimates of intracellular concentration from our lysate assay experiments (Fig. 2e), though they are roughly two orders of magnitude higher. This may reflect potential noise in the model or true biological variation, as the deep mutational scanning experiment was performed in a turbidostat maintained at low optical density and rapid exponential growth rather than in small-volume batch cultures. Again, we computed epistasis of each DHFR mutation in the Q33S and R166Q TYMS contexts relative to the WT TYMS background. Overall both the predicted growth rates and pattern of epistasis showed good

agreement to our experimental observations (Fig. 5a, b). The strong correlation ($R^2 = 0.84$) between the experimental growth rate data and the model across hundreds of in vitro characterized DHFR/TYMS genotypes indicated that our relatively simple two-equation model–which fits only a single parameter for DHFR enzyme abundance across all mutant variants–allows for reasonable growth rate predictions.

Having established model performance on this subset of 114 biochemically characterized DHFR and TYMS sequences, we next examined consistency of the model with all growth rate measurements (the total model fit). However the effect of most mutations on catalysis is unknown. Thus, for each DHFR point mutant we used Monte Carlo sampling to identify a space of $k_{cat}$ and $K_m$ values consistent with the three growth rate measurements (in the three TYMS backgrounds). While three growth rate measurements were insufficient to uniquely constrain both $k_{cat}$ and $K_m$ (the solution space is degenerate), this process did permit estimation of $\log_{10}$ catalytic power ($k_{cat}/K_m$) for all 2696 characterized point mutants. For the subset of biochemically characterized DHFR mutants we

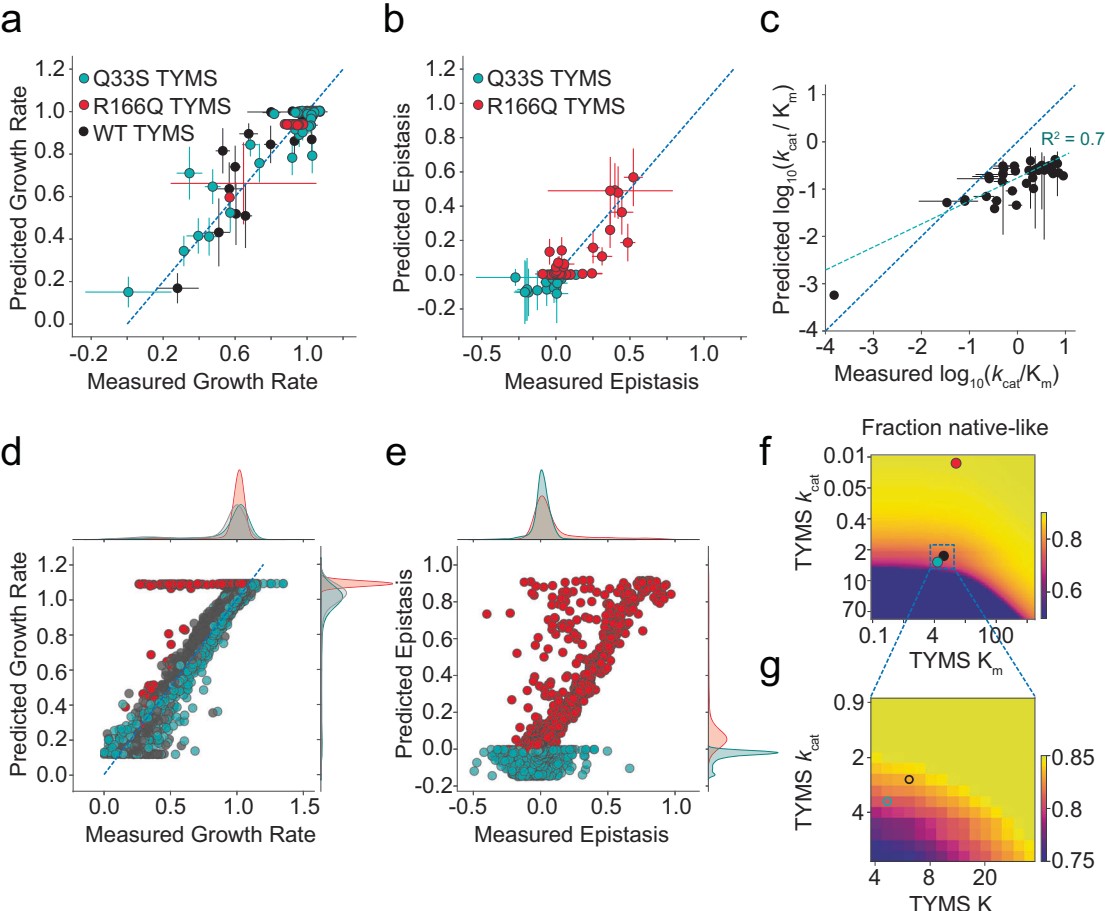

**Fig. 5 | Global comparison of the biochemistry-to-growth model and deep mutational scanning data set. a** Correlation between the experimentally measured and predicted growth rates of 114 DHFR/TYMS mutant combinations. Horizontal error bars indicate standard deviation of three replicate experimental growth rate measurements, vertical error bars are the standard deviation in predicted growth rates estimated from 1000 bootstrap re-samplings (and model fits) of the data (points are mean-centered). **b** Correlation between the experimentally measured and model-predicted epistasis, as computed from the growth rate data in (**a**). Again, color coding indicates TYMS background (identical to **a**). The dashed blue line indicates y = x. Horizontal error bars indicate standard deviation propagated from the experimentally measured growth rates across three replicate measurements (centered at the mean), vertical error bars are the standard deviation in the predicted epistasis values estimated by performing 1000 bootstrap re-samplings (and model fits) of the data (again centered at the mean). **c** Correlation between experimentally measured and computationally inferred $\log_{10}(k_{cat}/K_m)$ values for 38 mutants of DHFR. Horizontal error bars describe the standard deviation across triplicate experimental measurements, vertical error bars indicate the standard deviation across 50 iterations of stochastic (Monte-Carlo based) model inference. The circle markers are centered at the mean. **d** Correlation between experimentally measured and predicted growth rates across the entire deep mutational scanning dataset. The marginal distribution of growth rate effects is shown along each axis. **e** Correlation between experimentally measured and predicted epistasis across the entire deep mutational scanning dataset. The marginal distribution of epistatic effects is shown along each axis. **f** Mutational tolerance of DHFR as a function of TYMS background. The heatmap shows the fraction of DHFR mutations with relative growth rates of 0.9 or better as TYMS $k_{cat}$ and $K_m$ are discretely varied. Both axes are natural log spaced, TYMS $k_{cat}$ was sampled at 50 points between ln(-2) and ln(2), while TYMS $K_m$ was sampled at 50 points between ln(−1) and ln(3). The values for TYMS R166Q, Q33S and WT are marked with red, cyan and black circles respectively. **g** A zoomed-in version of (**f**), focusing on the mutational tolerance of DHFR for TYMS backgrounds similar in velocity to WT and Q33S TYMS.

observed reasonable agreement between the in vitro measurements and those inferred from our experimental growth rate data ($R^2 = 0.74$, Fig. 5c). In the current version of our model, mutant-specific changes in DHFR abundance will be collapsed into the $k_{cat}$ parameter. This results in an effective measure of $k_{cat}$ that captures both changes in catalytic activity and abundance, introducing one potential source of inaccuracy in the inference of catalytic power. Nonetheless, we find our relatively few-parameter model allows for reasonable inference of fold changes in catalytic power.

Once catalytic parameters were estimated across all point mutants, we put them back into the model to assess the correspondence between the predicted (modeled) growth rates, predicted epistasis, and our experimental observations, yielding a global picture of model fit quality. Overall, we observed that the model well-described the data with two exceptions. First, there was a small proportion of DHFR mutations that were predicted to be rescued by TYMS R166Q but in actuality were not (70 total, 2% of all DHFR mutations, the horizontal stripe of red dots in Fig. 5d). It is possible that these mutations caused a growth rate defect through toxicity linked to DHFR mis-folding and aggregation, a mechanism not captured by our model. Alternatively, these mutations may have had differential effects on protein abundance. Second, there was a proportion of DHFR mutations predicted to have negative epistasis to TYMS R166Q but observed to exhibit mild positive epistasis (Fig. 5e). Again, these differences may be related to the fact that DHFR abundance is modeled with a single parameter across all mutants. These discrepancies between true and effective $k_{cat}$ could be addressed in future work by including additional high-throughput assay data on stability or abundance to tease apart mutational effects on catalysis from stability and abundance[36].

Nevertheless, the data indicated that our model can globally describe growth rate phenotypes given variation in enzyme velocity.

The resulting model and inferred catalytic parameters now permit estimation of DHFR single mutant fitness in any TYMS background. We computed the fraction of DHFR point mutants that are neutral (growth rate above 0.9) as a function of variation in TYMS $k_{cat}$ and $K_m$. These calculations highlighted that selection on DHFR activity is strongly shaped by TYMS background, with low-activity TYMS variants increasing the mutational tolerance of DHFR (Fig. 5f, g). This suggests that TYMS inhibition or loss of function could promote the evolvability of DHFR both in the clinic and laboratory settings.

## DHFR/TYMS epistasis is organized into structurally localized groups

Next, we examined the structural pattern of DHFR positions with epistasis to TYMS Q33S and TYMS R166Q. Given that mutations tend to have similar epistatic effects at a particular DHFR position in our data set (Supplementary Fig. 4), we used k-means clustering to sort

positions into four categories according to their pattern of epistasis: negative, insignificant, positive, and strong positive (Fig. 6a, Supplementary Table 4). The strong positive category solely contained DHFR mutations in the TYMS R166Q background, while the negative epistasis category was predominantly occupied by DHFR mutations in the TYMS Q33S background. Based on our mathematical model, we expect that positions with strong positive epistasis to TYMS R166Q will have large deleterious effects on both $k_{cat}$ and/or $K_m$. Likewise, we expect that mutations with negative epistasis to TYMS Q33S should reduce $k_{cat}$ and/or increase $K_m$. Consistent with expectation, mapping the strongly epistatic positions to the DHFR structure showed that epistasis is organized into spatially distinct regions of the tertiary structure known to play a key role in catalytic function (Fig. 6b, c). Mutations with negative epistasis to Q33S tended to be proximal to the DHFR active site, particularly the folate binding pocket. The negative epistasis cluster included several key positions near or in the Met-20 loop, which is known to undergo conformational fluctuations associated with catalysis (residues A9,

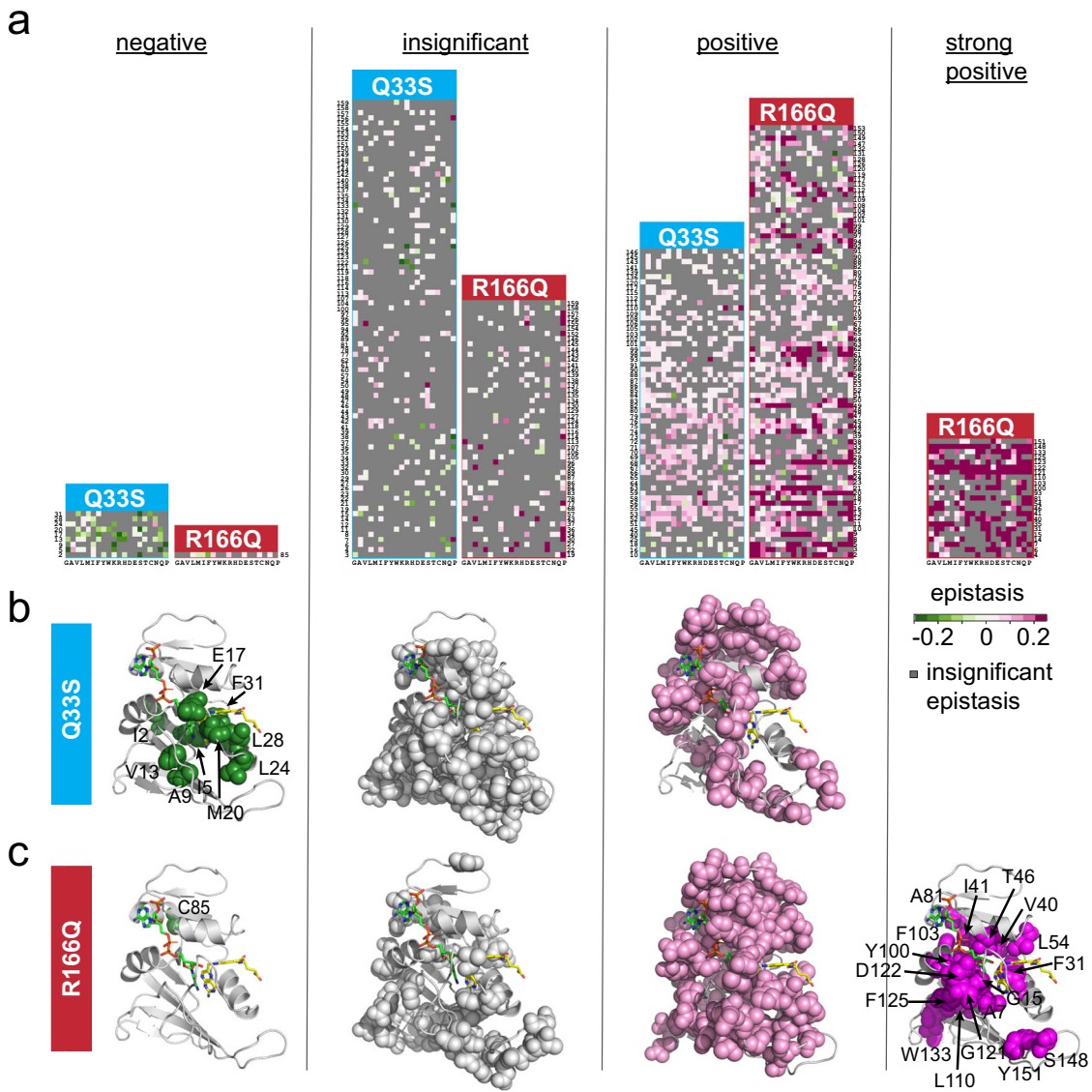

**Fig. 6 | DHFR positions clustered by epistatic mutational effect. a** Clusters of DHFR positions organized by predominant epistasis type. In each heat map DHFR positions are ordered along the vertical axis; amino acid residues are organized by physiochemical similarity along the horizontal axis. As in earlier plots, green indicates negative epistasis, and pink indicates positive epistasis. Gray pixels mark mutations with statistically insignificant epistasis. **b** Structural location of epistatic clusters for DHFR to TYMS Q33S. The DHFR backbone is in gray cartoon (PDBID: 1RX2[32]). Folate, the DHFR substrate is indicated with yellow sticks. The NADP+ cofactor is in green sticks. **c** Structural location of epistatic clusters for DHFR to TYMS R166Q. Color coding is identical to panel (**b**).

V13, E17 and M20)[29,32]. It also encompassed positions I5, L24, L28, and F31 which surround the folate substrate. Several of these positions have known roles in catalysis; mutations at position 31 promoted product release (while slowing hydride transfer), and dynamics of the M20 loop (which includes V13,E17) are associated with substrate binding and product release[33,37]. Additionally, specific mutations at positions 5, 20, and 28 result in trimethoprim resistance by altering trimethoprim affinity[37]. These structural and biochemical observations are consistent with the finding that mutations with negative epistasis tended to yield moderate to severe growth rate defects. Given their structural location near the folate binding pocket, it is possible that some of these negatively epistatic mutations reduce affinity (and increase $K_m$) for dihydrofolate substrate. For example, prior work has shown that mutations at position 28 increase $K_m$[38]. In contrast, positions with positive epistasis to Q33S often had very little (or sometimes a beneficial) effect on growth rate, and were distributed around the DHFR surface (Fig. 4c, Fig. 6b). In the context of TYMS R166Q only one position—C85—was included in the negative epistasis cluster (Fig. 6c). A large fraction of DHFR positions (53%, 84 total) displayed positive epistasis to TYMS R166Q; these positions were distributed throughout the DHFR structure. The positions in the strong positive epistasis cluster included mutations with some of the most severe effects on growth rate in the WT TYMS context. A number of these positions were previously established as important to DHFR catalysis, including residues F31, L54, G121, D122, and S148[29]. Mutations at these sites can be detrimental to $k_{cat}$, $K_m$, or both. The finding that highly epistatic mutations are concentrated at positions associated with DHFR catalysis provides additional support for the model that DHFR/TYMS coupling is mediated by a shared constraint on relative enzymatic activity.

## Epistasis and the structural encoding of DHFR catalysis

When the epistatic clusters are viewed together on the structure, one sees that they form approximate distance-dependent shells around the active site (Fig. 7a–d). Considering the pattern of epistasis to TYMS Q33S, positions with negative epistasis were closest to the active site, surrounded by positions with insignificant epistasis, and finally positions in the positive epistasis cluster form an outer shell (Fig. 7a, b). For TYMS R166Q, positions in the strong positive epistasis cluster were closest to the active site, followed by positive epistasis positions, and finally those with insignificant epistasis (Fig. 7c, d). For comparison, we also mapped the model-predicted catalytic power averaged across all mutations at a position to the structure (Fig. 7e). Together, these structural images paint a picture of the molecular encoding of catalysis and epistasis. Mutations with predicted intermediate-to-large effects on catalysis were nestled near the active site and showed negative epistasis to Q33S and strong positive to positive epistasis to R166Q. Mutations with more mild effects on catalysis showed weaker positive to insignificant epistasis to R166Q and Q33S. Though catalysis and epistasis showed an approximate distance-dependent relationship to the DHFR active site, there a number of key positions distal to the active site that exhibited large growth rate effects, strong positive epistasis to TYMS R166Q, and likely act allosterically to tune catalytic activity (e.g. L110, G121, D122, W133, S148, and Y151). The positions with the largest estimated effects on catalysis were highly evolutionarily conserved ($P < 10^{-10}$ by Fisher's exact test, Supplementary Table 5, Fig. 7f), indicating that our model and experimental data are capturing features relevant to the fitness of DHFR. Together, these data show that TYMS metabolic context strongly modulates the constraints on DHFR activity and catalysis.

## Discussion

It is well-appreciated that physical protein interactions place constraints on the individual interacting monomers. Protein interfaces are organized to bind with appropriate affinity and avoid non-specific interactions[39,40]. The individual components of physical complexes tend to be expressed in similar ratios to avoid dosage related toxicity and aggregation[41,42]. However the extent to which interactions mediated by biochemistry (rather than binding) constrain the function and sequence of individual monomers has remained less clear. We have explicitly revealed these interactions at single-residue resolution for one model system and coupled them with a mathematical model to quantify the intracellular constraints on DHFR and TYMS relative catalytic activities. While this study focuses on all possible DHFR single mutations in the context of a few TYMS variants, it would be reasonably straightforward to design analogous experiments that more densely sample TYMS variation, or quantify amino acid resolution epistasis to other folate metabolic enzymes. While practical constraints on sequencing depth make it difficult to imagine extending these experiments to cover all possible DHFR/TYMS double mutations, one could design targeted libraries that sample variation at evolutionarily conserved and/or catalytically important positions.

Our mutagenesis data and modeling show that TYMS activity strongly modifies the constraints on DHFR catalytic parameters; shaping both the range and relative importance of $k_{cat}$ and $K_m$ in modulating growth. This biochemical interaction results in an approximately shell-like pattern of mutational sensitivity to TYMS background (epistasis) in the DHFR tertiary structure. Extreme loss-of-TYMS function rescued strongly deleterious mutations in some of the most conserved DHFR active site positions, while moderate TYMS loss-of-function rescued moderately deleterious or weakly deleterious mutations at more peripheral solvent exposed sites. Given these data, we anticipate that inhibition or loss-of-function in TYMS could promote the evolvability of DHFR in nutrient rich environments by reducing the constraints on DHFR sequence and activity, a hypothesis with consequences for both laboratory and clinical evolution. For example, inhibiting TYMS activity in the clinic may promote the evolution of drug resistance in DHFR, while activating TYMS may restrict evolutionary accessible paths. We note that TYMS loss-of-function variants seem to be viable in some natural contexts, as prior work has identified trimethoprim-resistant clinical isolates with thymidine auxotrophy[19]. In the laboratory, strains with reduced TYMS activity could provide a less stringent context for testing designed sequences or evolving new DHFR function.

The existence of an enzyme velocity to growth-rate mapping—by definition—allows us to relate variation in DHFR and TYMS catalytic parameters to growth rate. It also allows one (in principle) to do the inverse: infer in vitro catalytic parameters from growth rate measurements. The intuition follows from classic steady-state Michaelis Menten experiments: to quantify steady state kinetics in vitro one measures enzyme initial velocity as a function of substrate concentration. In our sequencing-based experiments, variation in TYMS background effectively titrates intracellular concentrations of DHF (substrate) while growth rate provides an estimate of velocity. Though our current dataset of three TYMS backgrounds is insufficient to uniquely constrain precise fits for $k_{cat}$ and $K_m$, we anticipate that the addition of a few additional TYMS backgrounds and/or the use of more sophisticated fitting approaches will permit more accurate biochemical parameter inference. Indeed, recent work on the small peptide binding proteins (the PDZ and SH3 domains) has shown how measuring the growth rate effect of mutations in different genetic backgrounds and assay conditions can well-constrain biophysical parameters for binding affinity and protein stability[36,43]. One might follow a conceptually similar strategy to learn quantitative biochemical parameters from high throughput growth rate data. New microfluidics-based approaches for high-throughput biochemistry could play a key role in refining and testing such methodology[44].

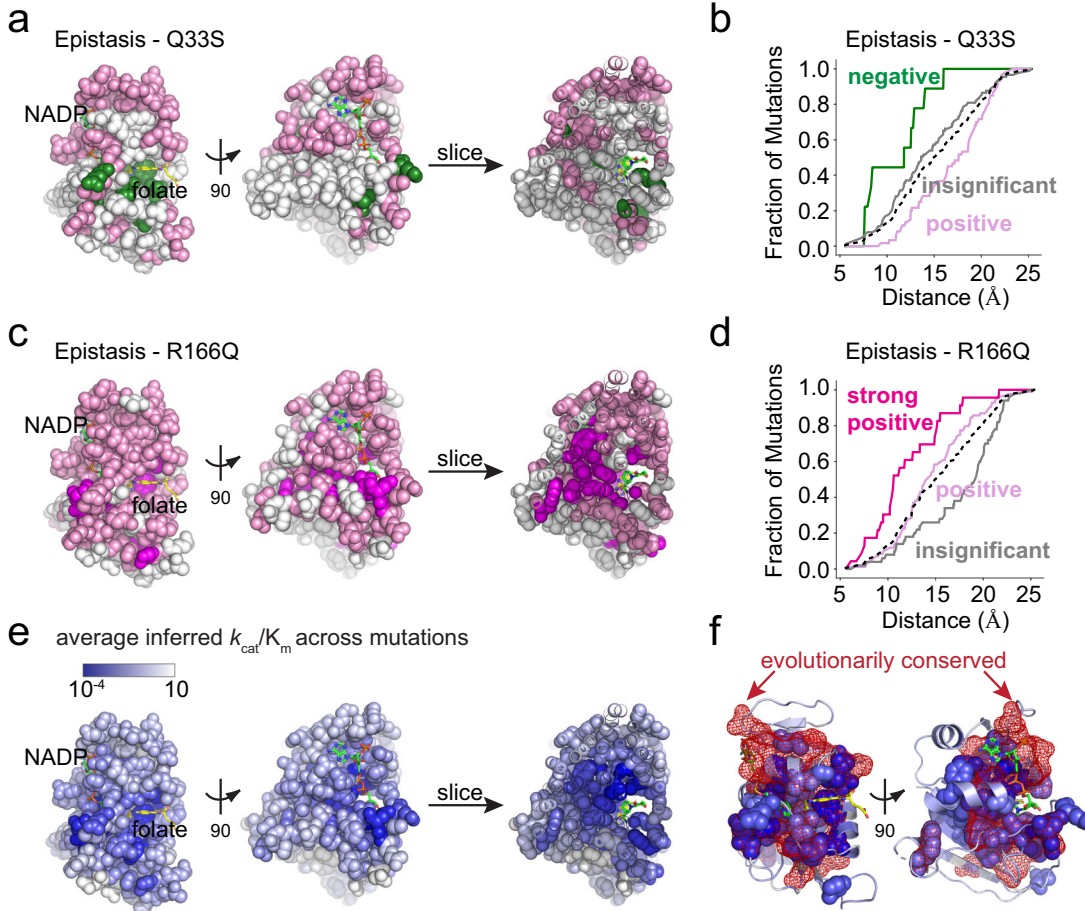

**Fig. 7 | The structural organization of epistasis in DHFR. a** Epistasis of DHFR positions to TYMS Q33S. The DHFR structure is in space-filling spheres (PDBID: 1RX2), NADP co-factor in green sticks, and folate in yellow sticks. A slice through the structure shows the interior arrangement of epistasis. Positions in the negative epistasis cluster are green, positions in the positive epistasis cluster are pink. Gray spheres indicate the insignificant epistasis cluster. **b** Cumulative distribution of positions in each epistatic cluster by distance to the DHFR active site for the TYMS Q33S background. In this case, active site was defined as the C6 atom of folate. Color coding follows from (**a**), the black dashed line indicates the distance distribution of all residues in the protein. **c** Epistasis of individual DHFR positions to TYMS R166Q. Color coding follows from (**a**) with the addition that positions with strong positive epistasis are magenta. **d** Cumulative distribution of positions in each epistatic cluster by distance to the DHFR active site in the TYMS R166Q background. In this case, active site was defined as the C6 atom of folate. Color coding follows from (**c**), the black dashed line indicates the distance distribution of all residues in the protein. **e** The position-averaged $\log_{10}$ catalytic power (as inferred from the growth rate data) across measured mutations. All residues are indicated in space filling and color coded by the average mutational effect. Blue indicates positions where mutations have a deleterious effect on catalytic power (on average), while white indicates mutations that have little effect on catalytic power. The NADP co-factor is shown in green sticks, and folate in yellow sticks. A slice through the structure shows the interior distribution of mutational effects on catalysis. **f** Structural overlap between positions associated to catalysis and evolutionary conservation. The DHFR backbone is in gray cartoon, the NADP co-factor in green sticks, and folate in yellow sticks. Positions where mutations have a deleterious effect on catalysis (at least half a standard deviation below the mean) are in blue space filling. Evolutionarily conserved positions (as computed by the Kullback-Leibler relative entropy in a large alignment of DHFR sequences) are in red mesh.

Together our findings shape how we think about designing enzymes and metabolic systems. Typical strategies for designing enzymes do not explicitly consider cellular context[45]. As a result, a significant fraction of designs could fail simply because they are not properly matched in terms of velocity to the surrounding pathway. The limited ability of homologs to complement growth in another species has been observed for a number of enzymes[46–50], including DHFR[51,52]. Thus, even a well-designed catalytically active synthetic enzyme could fail to rescue growth if placed in the wrong cellular context. Just as computational protein design considers entire physical complexes to create binding interactions with altered affinity and specificity, one might consider the joint design of biochemically-interacting enzymes to alter metabolic efficiency and growth. Further study of enzyme rates and abundance across species, as well as characterizations of enzyme velocity to growth rate mappings, will help shape our understanding of the system level constraints placed on metabolic enzymes.

## Methods

### Escherichia coli expression and selection strains

ER2566 Δ*folA* Δ*thyA E. coli* were used for all growth rate measurements; this strain was a kind gift from Dr. Steven Benkovic and is the same used in Reynolds et al. 2011 and Thompson et al.[38,53]. XL1-Blue *E. coli* (genotype: *recA1 endA1 gyrA96 thi-1 hsdR17 supE44 relA1 lac* [F' *proAB lacI^qZΔM15* Tn10(Tet^r)]) from Agilent Technologies were used for cloning, mutagenesis, and plasmid propagation. BL21(DE3) *E. coli* (genotype:*fhuA2 [lon] ompT gal (λ DE3) [dcm] ΔhsdS. λ DE3 = λ sBamHIo ΔEcoRI-B int::(lacI::PlacUV5::T7 gene1) i21 Δnin5*) from New England Biolabs were used for protein expression.

### Selection vector for DHFR constructs

DHFR variants were cloned into a modified version of the pACYC-Duet 1 vector (Novagen), which we refer to as pTet-Duet. pTet-Duet is a low-copy number vector containing two multiple cloning sites; the first is under control of the T7 promoter and the second was modified to be

regulated by the tetracycline repressor (TetR). DHFR (*folA*) is cloned into the first MCS; TYMS (*thyA*) is cloned into the second MCS. During selections we do not induce expression of either gene but instead rely on leaky expression in ER2566 Δ*folA* Δ*thyA E. coli*. We have deposited all of these constructs with Addgene alongside plasmid maps [https://www.addgene.org/browse/article/28229429/].

## Expression vector for DHFR constructs
*E.coli folA* (the gene encoding DHFR) was cloned into the pHis8-3 expression vector using restriction sites NcoI and XhoI. DHFR was tagged in-frame with an N-terminal 8X-Histidine tag separated from the *folA* reading frame by a thrombin cleavage site. Individual point mutant clones were constructed using the Quikchange II site-directed mutagenesis kit (Agilent).

## Expression vector for TYMS constructs
The *thyA* gene (encoding TYMS) was amplified by PCR from *E. coli* MG1655 and cloned into the vector pET24A using XbaI/Xho restriction sites. The point mutants of TYMS (Q33S, R127A, and Q33S) were made using the Agilent QuikChange II site-directed mutagenesis kit. These constructs are available on AddGene [https://www.addgene.org/browse/article/28229429/].

## Plate-reader based growth rate assays
We constructed a plasmid-based series of DHFR point mutants (spanning a range of catalytic activities) in four TYMS backgrounds by quickchange mutagenesis. The sequence-verified mutants were deposited with Addgene ([https://www.addgene.org/browse/article/28229429/]). DHFR and TYMS point mutant combinations in the selection vector were transformed into ER2566 Δ*folA* Δ*thyA* chemically competent cells by heat shock. The cells were recovered for 60 min at 37 °C with shaking at 220 rpm, spread on agar plates (Luria Broth (LB) containing 30 μg/ml chloramphenicol and 50 μg/ml thymidine), and grown at 37 °C overnight. The next day, liquid overnight cultures were inoculated from a streak over multiple colonies and grown overnight at 37 °C in LB supplemented with 30 μg/ml chloramphenicol and 50 μg/ml thymidine. These overnight cultures were pelleted and washed with M9 minimal media, then resuspended in pre-warmed M9 media supplemented with 0.4% glucose, 0.2% amicase, 2 mM MgSO$_4$, 0.1 μM CaCl$_2$, 30 μg/ml chloramphenicol (henceforth referred to as M9 selection media). Next, OD$_{600}$ for all resuspended cultures was measured in a Perkin Elmer Victor X3 plate reader. Cultures were then diluted to OD$_{600}$ = 0.1 in prewarmed M9 selection media with 50 μg/ml thymidine and incubated for 4 h at 30 °C, shaking at 220 rpm. After this period of adaptation and regrowth, cultures were back-diluted to OD$_{600}$ = 0.1 in 1 ml prewarmed M9 selection media with 50 μg/ml thymidine. These cells were inoculated into 96-well culture plate at OD$_{600}$ = 0.005 (10 μl cells into 200 μl total well volume) containing prewarmed M9 selection media with 50 μg/ml thymidine; plates were sealed with EasySeal permeable covers (Sigma Aldrich). All growth rate measurements were made in triplicate. Plates were shaken for 10 seconds before reading, and Readings of OD$_{600}$ were taken every 6 min over 24 h using a BioTek Synergy Neo2 plate reader in a 30 °C climate-controlled room.

## DHFR saturation mutagenesis library construction
The DHFR saturation mutagenesis library was constructed as four sublibraries in the pTet-Duet selection vector (see above for selection vector details). Each sublibrary combines mutations within 40 contiguous amino acid positions to ensure that the mutated region can be completely covered with short read sequencing (a 300 cycle v2 Illumina sequencing kit). The regions spanned by each sublibrary were as follows: amino acid positions 1-40 (sublibrary 1, SL1), 41-80 (sublibrary 2, SL2), 81-120 (sublibrary 3, SL3), and 121-159 (sublibrary 4, SL4). 'Round the Horn' or inverse

PCR (iPCR) with mutagenic NNS primers (N = A/T/G/C, S = G/C) was used to introduce all 20 amino acid substitutions at a single amino acid position as described in Thompson et al.[38] (see Appendix 1 of Thompson et al for a complete list of primers). Library completeness was verified by deep sequencing. In our initial validation sequencing run we found that mutations at positions W22 and L104 were systematically under-represented; iPCR was repeated for these positions and they were supplemented into their respective assembled sublibraries.

After sub-library assembly, restriction digest and ligation were used to subclone each sublibrary into pTet-Duet plasmids containing the three different TYMS backgrounds (WT, R166Q, or Q33S). The entire DHFR coding region containing restriction sites (NotI and EcoNI) was amplified by PCR. PCR reaction was size-verified with agarose gel electrophoresis with an expected band size of 627 bp. The library PCR products and target plasmids were double digested with NotI and EcoNI for 3 h at 37 °C. To prevent re-circularization, the digested plasmid was treated with Antarctic phosphatase for 1 h at 37 °C. The DHFR insert and treated plasmid were ligated with T4 DNA ligase overnight at 16 °C. The concentrated ligation product was then transformed into *E. coli* XL1-blue by electroporation, and recovered in SOB for 1 h at 37 °C. 20 μL of the recovery culture was serially diluted and plated on LB-agar with 50 μg/mL thymidine and 30 μg/mL chloramphenicol, to permit quantification of transformants and estimate library coverage following ligation into the alternate TYMS backgrounds. The minimum library coverage was 1000 colony forming units (CFU) per mutation. The remaining recovery culture was grown in a flask containing 12 ml LB with 30 μg/mL chloramphenicol and 50 μg/mL thymidine at 37 °C, with 220 rpm shaking overnight. 10 ml of the overnight culture was miniprepped with the Gene-Jet Mini-prep kit (Fisher Scientific, K0503) to obtain the plasmid library. The DHFR deep mutational scanning libraries (in all three TYMS backgrounds) have been deposited at Addgene ([https://www.addgene.org/pooled-library/reynolds-dhfr-mutagenesis], under Addgene IDs 1000000194, 1000000195,1000000196).

## Growth Rate Measurements in the Turbidostat for all DHFR mutant libraries
All sublibraries were inoculated, grown, and sampled in triplicate. Each plasmid sublibrary was transformed into the *E. coli* double knockout strain ER2566 Δ*folA* Δ*thyA* by electroporation and recovered in SOB for one hour at 37 °C. At this transformation step we again estimated library coverage to ensure the complete library was transformed into our selection strain. To estimate library coverage, 20 μL of the recovery culture was serial diluted with SOB and plated on LB agar plates containing 30 μg/mL chloramphenicol and 50 μg/mL of thymidine. The remainder of the recovery culture was inoculated into M9 selection media supplemented with 50 μg/mL thymidine and grown overnight at 37 °C. All selection experiments in this work had an estimated library coverage of 1000 CFU/mutant or greater. The overnight liquid culture was washed and back-diluted to OD$_{600}$ = 0.1 in M9 selection media supplemented with 50 μg/mL thymidine, and incubated for four hours at 30 °C to allow adaptation to selection temperature and to return the culture to log-phase growth. Following adaptation, selection was initiated by back-diluting these cultures to an OD$_{600}$ of 0.1 into 17 mL of pre-warmed M9 selection media supplemented with 50 μg/mL thymidine in continuous culture vials. These vials were then incubated in a turbidostat with a target OD$_{600}$ of 0.15 at a temperature of 30 °C. The turbidostat maintained a set optical density by adding 2.8 mL dilutions of M9 selection media supplemented with 50 μg/mL thymidine in response to OD detection, and was built according to the design of Toprak et al.[54] Culture samples (1 mL each) were taken at the beginning of selection (t = 0 hours) and at 4, 8, 12, 20, and 24 h into selection. Immediately after each time point, these 1 mL samples were pelleted by centrifugation, supernatant removed, and stored at −20 °C.

## Next generation sequencing amplicon sample preparation

Each turbidostat selection sample (representing a particular time-point for a sublibrary and replicate) was prepared for sequencing as a PCR amplicon using Illumina TruSeq-HT i5 and i7 indexing barcodes. To generate these amplicons, each cell pellet from the growth rate assay was thawed and lysed by resuspending the cells with 100 μL dH$_2$O and incubation at 95 °C for 5 min. Lysates were then clarified by centrifugation at maximum speed for 10 min in a room temperature bench top microcentrifuge. Supernatants containing plasmids were isolated from the pellet. 1 μL of each supernatant was used as the template for an initial round of PCR with Q5-Hot Start Polymerase (NEB) that amplified the DHFR coding region of the sublibrary (10 PCR cycles total, standard Q5 reaction conditions). From this first PCR reaction, 1 μL was used in a second round of PCR (22 cycles of denaturation/anneal/elongation) with primers that added Illumina sequencing adaptors. The sequencing primers are specified in Supplementary Table 6. Together, these two rounds of PCR yielded a final product of size: 315 bp (SL1), 308 bp (SL2), 298 bp (SL3), 304 bp (SL4). The amplicons were size verified using 1% agarose gel electrophoresis. In the case where a sample did not produce an amplicon, the first round PCR was repeated with 2 μL of the supernatant rather than 1 μL, with the remaining preparation identical. All amplicons were individually quantified using with Quant-iT™ PicoGreen™ dsDNA Assay Kit (ThermoFisher Scientific) and mixed in equimolar ratio, with a final target amount greater than or equal to 2000 ng. Errors in pipetting volume were minimized by ensuring that more than 2 μL was taken from each amplicon. This mixture was gel-purified and then cleaned and concentrated using the Zymo Research DNA Clean & Concentrator-5 kit. Purity was assessed by A260/A80 and A260/A230 nm absorbance ratios. The sample library DNA concentration was measured using a Qubit dsDNA HS Assay in a Qubit 3 Fluorometer (Invitrogen by Thermo-Fisher Scientific). The sample library was diluted to 30 nM in a volume of 50 μL of TE buffer (1 mM Tris-HCl (pH 8.5), 10 mM EDTA (pH 4)). This mixed and quantified library was sequenced on an Illumina HiSeq (150 cycle x 2 paired-end) by GeneWiz. The NGS sequencing run resulted in 252 GB of data, consisting of 337,353,664 reads. The raw data have been deposited with the NCBI sequencing read archive under project identifier PRJNA791680.

## DHFR expression and purification

DHFR mutant variants were expressed in BL21(DE3) *E. coli* grown at 30 °C in 50 ml Terrific Broth (TB) with 35 μg/ml Kanamycin (Kan) for selection. Expression was induced at an OD$_{600}$ = 0.6−0.8 with 250 uM IPTG, and cells were grown at 18 °C for 16−18 h. Cultures were pelleted by centrifugation for 10 min at 5000 × *g*, 4 °C and supernatant removed; cell pellets were stored at −80 °C. Thawed cell pellets were lysed by sonication in 10 ml lysis buffer (50 mM Tris, 500 mM NaCl, 10 mM imidazole, pH 8.0 buffer containing 0.1 mM PMSF, 0.001 mg/ml pepstatin, 0.01 mg/ml leupeptin, 20 μg/ml DNAseI and 5 μg/mL lysozyme). The resulting lysate was clarified by centrifugation and incubated with 0.1 ml Ni-NTA agarose (Qiagen) slurry (0.05 ml column volume) equilibrated in Nickel Binding Buffer (NiBB, 50 mM Tris pH 8.0, 500 mM NaCl, 10 mM imidazole) for 15 min on a tube rocker at 4 °C. The slurry was then transferred to a disposable polypropylene column (BioRad). After washing with 10 column volumes (CV) of NiBB, DHFR was eluted with 0.5 mL 50 mM Tris pH 8.0, 500 mM NaCl, 400 mM imidazole. The eluted protein was concentrated and buffer-exchanged to 50 mM Tris, pH 8.0 in a 10 kDa Amicon centrifugal concentrator (Millipore) and centrifuged 15 min at 21,000 × *g*, 4 °C to pellet any precipitates. Following buffer exchange, the protein was purified by anion exchange chromatography (using a BioRad HiTrapQ HP column on a BioRad NGC Quest FPLC). A linear gradient was run from 0 to 1 M NaCl in 50 mM Tris pH 8.0 over 30 ml (30 column volumes, CV) while collecting 0.5 ml fractions. Fractions containing

DHFR were combined, concentrated, flash-frozen in liquid nitrogen, and stored at −80 °C.

## TYMS expression and purification

Individual TYMS mutants were expressed in BL21(DE3) *E. coli* grown at 37 °C in 50 ml Terrific Broth (TB) with 35 μg/ml Kanamycin (Kan) for selection. Expression was induced with 1 mM IPTG when the cells reached an OD$_{600}$ = 0.6−0.8, and the cells were then grown at 18 °C for 16−18 h before harvesting pellets for storage at −80 °C. TYMS was purified from the frozen pellets following a protocol adapted from Changchien et al.[55] Cell pellets were thawed and resuspended in TYMS lysis buffer (20 mM Tris, 10 mM MgCl2, 0.1% deoxycholic acid, pH 7.5 with 5 mM DTT, 0.2 mg/ml lysozyme, 5 μg/ml DNAse I) and incubated at room temperature while rocking for 15 min. The resulting supernatant was clarified by centrifugation. Next, strepto-mycin sulfate was added to a final concentration of 0.75% to separate nucleic acids. The cells were incubated rocking at 4 °C for 10 min and the supernatant was retained following centrifugation for 10 min at >10,000 × *g*. Ammonium sulfate was then added at 50% saturation (0.3 g/ml), mixed for 10 min at 4 °C, then centrifuged as above, retaining supernatant. Additional ammonium sulfate was then added to the supernatant at 80% saturation (an additional 0.2 g/ml), mixed for 10 min at 4 °C, and centrifuged as above, retaining the pellet. The pellet was dissolved in 25 mM potassium phosphate pH 6.5 and dialyzed overnight at 4 °C against 1 L 25 mM potassium phosphate pH 6.5. Following dialysis the protein was purified by anion exchange (HiTrap Q HP column, Cytiva) with a 25 CV linear gradient from 0 M NaCl to 1 M NaCl in 25 mM potassium phosphate pH 6.5. FPLC fractions containing TYMS were combined and concentrated using a 10 kDa Amicon concentrator (Millipore) and stored at 4 °C for up to a week.

## DHFR steady-state Michaelis Menten kinetics

DHFR $k_{cat}$ and K$_m$ were determined under Michaelis-Menten conditions with saturating concentrations of NADPH as in prior work[53,56]. Briefly, DHFR protein concentration was determined by measuring A$_{280}$ (extinction coefficient = 33500 M$^{-1}$cm$^{-1}$). DHF (Sigma Aldrich) was prepared in MTEN buffer (50 mM MES, 25 mM Tris base, 25 mM Ethanolamine, 100 mM NaCl, pH 7.0) containing 5 mM DTT (Sigma Aldrich). 100 nM DHFR protein and 100 μM NADPH (Sigma Aldrich) were combined in MTEN buffer with 5 mM DTT and pre-incubated for 1 h at 25 °C prior to measurement. To initiate the reaction, the protein-NADPH solution was mixed with DHF in a quartz cuvette (sampling DHF over a range of concentrations, tuned to the Km of the mutant). The initial velocity of DHFR was measured spectro-photometrically by monitoring the consumption of NADPH and DHF (decrease in absorbance at 340 nm, $\Delta\varepsilon_{340}$ = 13.2 mM$^{-1}$ cm$^{-1}$). All measurements were made in triplicate; analysis was performed using the Michaelis-Menten nonlinear regression function of Graph Pad Prism.

## Preparation of TYMS substrate for assaying enzyme activity and steady state kinetics

(6 R)-methylenetetrahydrofolic acid (MTHF, CH$_2$H$_4$fol) was purchased from Merck & Cie (Switzerland) and dissolved to 100 mM in nitrogen-sparged citrate-ascorbate buffer (10 mM ascorbic acid, 8.5 mM sodium citrate, pH 8.0). 30 μL aliquots were made in light-safe microcentrifuge tubes, flash-frozen in liquid nitrogen, and stored at −80 C. Before use, the stock was thawed and diluted to 10 mM in TYMS kinetic reaction buffer (100 mM Tris base, 5 mM Formaldehyde, 1 mM EDTA, pH 7.5) and quantified in an enzymatic assay: 50 μM MTHF, 200 μM dUMP and 1 μM TYMS protein were combined and A$_{340}$ measured until steady-state reached. Actual concentration was then calculated from the difference in A$_{340}$ before and after the reaction using Beer's Law (MTHF extinction coefficient: 6.4 mM$^{-1}$cm$^{-1}$).

## TYMS lysate assays for estimation of intracellular protein abundance

We measured the TYMS activity in crude lysates of six *E. coli* strains: (1) WT ER2566, (2) ER2566 Δ*folA* Δ*thyA*, and the ER2566 Δ*folA* Δ*thyA* strain transformed with a pTet-Duet selection vector harboring DHFR D27N matched with either (3) TYMS R166Q, (4) TYMS R127A, (5) TYMS Q33S, or (6) TYMS WT. Cultures of each were grown to mid-log phase ($OD_{600} = 0.4$–$0.6$) under media conditions identical to our selection (M9 selection media supplemented with $50\,\mu g/mL$ thymidine). The resulting culture was pelleted for 10 min at 5000 x g, liquid media was decanted, and pellets were weighed and frozen at $-80\,°C$. We found that pellets from 250 ml of culture were sufficient for triplicate assays. Pellets were thawed and lysed with B-PERII reagent (Thermo Fisher) supplemented with 50 mM MgCl$_2$, 5 mM DTT, 0.2 mg/mL lysozyme, and 0.005 mg/mL DNAseI. The lysis volume was normalized to 1 ml per 0.1 mg of pellet weight. Once resuspended, all pellets were incubated in lysis reagent for 15 min at room temperature on a tube rotator, then pelleted at $21,000 \times g$ at $4\,°C$. The soluble fraction resulting from lysis was used for lysate based enzyme activity measurements.

To measure enzyme velocity, $720\,\mu L$ of soluble cell lysate was combined with $80\,\mu l$ of the TYMS substrates at saturating concentration ($150\,\mu M$ CH$_2$H$_4$fol and $100\,\mu M$ dUMP in TYMS assay buffer with 5 mM DTT) in a cuvette. Immediately following substrate addition, DHF accumulation was monitored spectrophotometrically by absorbance at 340 nM for 900 seconds. All measurements were referenced to a blank cuvette containing lysate with no addition of substrates. DHF concentration was calculated from $A_{340}$ using the DHF extinction coefficient ($\Delta\varepsilon_{340} = 6.4\,mM^{-1}\,cm^{-1}$) and Beer-Lambert equation. The resulting DHF concentration measurements over time were entered into GraphPad Prism and background-subtracted to zero for the initial time point across all samples. Enzyme velocity was estimated by linear fit of the first 200 seconds of data. Dividing this velocity by the $k_{cat}$ values for TYMS (as previously measured in our lab) yielded an estimate of intracellular TYMS concentration for each variant.

## TYMS steady-state Michaelis Menten kinetics

TYMS $k_{cat}$ and $K_m$ were determined for both dUMP and MTHF under Michaelis-Menten conditions by varying one substrate and holding the other saturating as in prior work[57,58]. Briefly, TYMS protein concentration was determined by measuring $A_{280}$ (extinction coefficient = $53400\,M^{-1}cm^{-1}$). TYMS protein was prepared in TYMS assay buffer (100 mM Tris base, 5 mM Formaldehyde, 1 mM EDTA, pH 7.5) containing 50 mM DTT (Sigma Aldrich). 50 nM TYMS protein and either 100 μM dUMP (Sigma Aldrich) or 150 μM MTHF (Merck & Cie) were combined with varying concentrations of the other substrate to initiate the reaction. The production of DHF was monitored spectrophotometrically (increase in absorbance at 340 nm, $\Delta\varepsilon_{340} = 6.4\,mM^{-1}cm^{-1}$) for 2 min per reaction. All measurements were made in triplicate; analysis was performed using the Michaelis-Menten nonlinear regression function of Graph Pad Prism.

## Enzyme velocity to growth rate model construction and parameterization

For the purposes of modeling, we approximated DHFR and TYMS as a two-enzyme cycle in which DHFR produces THF and consumes DHF, and TYMS produces DHF and consumes THF. This abstraction ignores the different carbon-carrying THF species, instead collapsing them into a single reduced folate pool. This simplification allows us to construct an analytically solvable model for steady state THF concentration that we can then relate to growth (Eq. 3).

First, we fit the free parameters in the Goldbeter-Koshland equation ([DHFR], [TYMS$_{WT}$], [TYMS$_{R166Q}$], fol$_{tot}$) using a set of ten metabolomics measurements for the relative abundance of the 3-glutamate form of formyl THF as obtained in prior work[1]. These measurements were made for DHFR mutations G121V, F31Y/L54I, M42F/G121V, F31Y/

G121V and the WT in the background of WT TYMS and TYMS R166Q. So why this particular folate species? We noticed that the relative abundance of many of the reduced THF species in our data set was correlated, and chose formyl THF to model because the experimental data were less variable and showed a strong, monotonic relationship with cell growth. Then, we fit the free parameters in equation two ($g_{max}$, $g_{min}$, $K$, $n$) using a set of ten growth rate measurements for the same DHFR/TYMS mutation pairs. This process gave rise to the fits shown in Fig. 1. When assessing model performance against the larger set of TYMS variants (as in Fig. 2) we refit all parameters ($g_{max}$, $g_{min}$, $K$, $n$, [DHFR], [TYMS$_{WT}$], [TYMS$_{R166Q}$], [TYMS$_{Q33S}$], [TYMS$_{R127A}$], fol$_{tot}$) to the growth rate data only since we did not have metabolomics data for this larger set. All parameter fits were made in python using the least_squares fitting function of the scipy.optimize module[59]; the complete fitting process is documented in Jupyter notebook 1_KGmodel.ipynb in the associated github repository.

We assessed the model sensitivity to shuffling the data (Supplementary Fig. 1e-h) by randomly shuffling all catalytic parameters ($k_{cat}$, $K_m$) 50 times across DHFR and TYMS and computing an $R^2$ value and RMSD. We also assessed model sensitivity to subsampling the data; error bars in Figs. 1, 2, and Supplementary Fig. 1g,h correspond to SEM across jackknife re-samplings of the data wherein one DHFR/TYMS combination was left out for each re-sampling. Finally, to assess the global model fit to the data (as in Fig. 5) we first fit the 9 model parameters ($g_{max}$, $g_{min}$, $K$, $n$, [DHFR], [TYMS$_{WT}$], [TYMS$_{R166Q}$], [TYMS$_{Q33S}$], fol$_{tot}$) using the growth rate measurements of 16 DHFR mutations for which experimental $k_{cat}$ and $K_m$ were known (48 total observations given the three TYMS backgrounds). Then, fixing these parameters, we fit $k_{cat}$ and $K_m$ values for all 2696 mutations with growth rate measurements in all three TYMS backgrounds to the complete data set of 8088 sequencing-based growth rate observations. This process is documented in Jupyter notebook 4_ModelAndDMSData.ipynb in the associated github repository.

## Next generation sequencing data processing and read counting

All PCR amplicons (corresponding to individual replicates, timepoints and sublibraries) were sequenced on an Illumina HiSeq using $2 \times 150$ paired end reads. The resulting fastq files were processed and filtered prior to read counting. Briefly, the forward and reverse reads were merged using USEARCH. Each read was quality score filtered (Q-Score $\geq 20$) and identified as a WT or mutant of DHFR using a custom python script. This python script filtered for full length reads and base call quality scores greater than 20 (error rate $\leq 1:100$). The reads passing these quality control criteria were compared against the wild-type reference sequence to determine mutation identify. Reads that contained multiple point mutations or mutations outside the sublibrary of interest were removed from analysis. This process resulted in counts for the WT and each mutant at each time point and replicate. These counts were further corrected given the expected error in the data (q-score) and Hamming distance from the WT codon to account for potential hopping of WT reads to mutations; a process that was detailed in McCormick et al.[56].

## Relative growth rate calculations

We calculated relative growth rates for individual mutations and the WT over time from the sequencing-based counts ($N_t^{mut}$, $N_t^{WT}$). Mutants with fewer than 10 counts were considered absent from the data set and were set to zero to reduce noise. From these thresholded counts, we calculated a log normalized relative frequency of each mutation over time:

$$\log_2(f(t)) = \log_2\left(N_t^{mut}/N_t^{WT}\right) - \log_2\left(N_{t=0}^{mut}/N_{t=0}^{WT}\right) \tag{5}$$

We then calculated relative growth rate ($m_{DHFmut}^{TS}$) as the slope of the log relative frequency over time by linear regression. Linear

regression was performed using scikit Learn, and individual points were weighted by the number of counts (in order to down weight less-sampled mutants at later time points). Relative growth rate of a mutant was only calculated if the mutant was present over at least the first three time points, otherwise it was classified as a Null mutant. Finally, all relative growth rates were normalized such that WT has a relative growth rate of 1. Growth rates were additionally normalized by the bulk culture growth rate (estimated from the turbidostat, in units of generations per hour) to account for small vial-to-vial variations culture doublings across the experiment. The standard error in growth rate was computed across triplicate measurements. All calculations are shown in Jupyter notebook 2_DMSGrowthRates.ipynb in the associated github repository.

### Epistasis analysis

Epistasis was calculated according to an additive model:

$$\varepsilon_{\text{DHmut,TSmut}} = m_{\text{DHmut}}^{\text{TS\_mut}} - m_{\text{DHmut}}^{\text{TS}_{\text{WT}}} \tag{6}$$

In our experiments TYMS R127A, Q33S and R166Q have no growth rate effect in the WT DHFR context due to thymidine supplementation. Under this formalism, mutations that show improved growth in the mutated TYMS background have positive epistasis, while mutations with reduced growth in the mutated TYMS background have negative epistasis. We assessed the statistical significance of epistasis by unequal variance $t$ test under the null hypothesis that the mutations have equal mean growth rates in both TYMS backgrounds (across three replicate measurements). These $p$ values were compared to a multiple-hypothesis testing adjusted $p$ value determined by Sequential Goodness of Fit ($P = 0.035$ for TYMS Q33S and $P = 0.029$ for TYMS R166Q)[35]. K-means clustering of epistatic positions was performed using a custom script based on that described in Thompson et al. [38] All epistasis calculations are shown in Jupyter notebook 3_Epistasis.ipynb in the associated github repository.

### Reporting summary

Further information on research design is available in the Nature Portfolio Reporting Summary linked to this article.

## Data availability

The raw sequencing data generated in this study were deposited in FASTQ format in the NCBI sequencing read archive, under BioProject ID PRJNA791680 [https://www.ncbi.nlm.nih.gov/bioproject/791680]. The processed growth rates and epistasis measurements (as inferred from the sequencing data) are available as Supplementary Data 1 and 2. All code used to process these data are available in github [release v1.0.0: https://doi.org/10.5281/zenodo.10845716]. The processed growth rates and epistasis measurements are also available in github (see Output directory) as tab-delimited text and python-importable pickle files. Metabolomics data for formyl THF and DHF used in model fitting were previously described[1] and are specified in the github python notebook 1_KGModel. Biochemical rate constants for DHFR and TYMS (compiled from both this study and other published works) used in model fitting can be found in Supplementary Tables 2 and 3. Parameter fits from all described iterations of model fitting are in Supplementary Table 1. The structural data for TYMS (1BID [https://doi.org/10.2210/pdb1BID/pdb]) and DHFR (1RX2 [https://doi.org/10.2210/pdb1RX2/pdb]) used in this study are available from the PDB.

## Code availability

Code for the enzyme velocity to growth rate model, and analysis of all deep mutational scanning data is available on github: https://github.com/reynoldsk/dhfr-tyms-epistasis. The stable DOI for the code release associated with this publication is: https://doi.org/10.5281/zenodo.10845716.

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

## Acknowledgements

We thank Olivier Rivoire for early conversations regarding our data, and Elliott Ross for critical feedback on both the modeling and manuscript. We also thank the Reynolds lab for feedback on experimental design, data analysis, and the manuscript. Research reported in this publication was supported by the National Institute of General Medical Sciences of the National Institutes of Health under Award Number R01GM136842 to KAR. The content is solely the responsibility of the authors and does not necessarily represent the official views of the National Institutes of Health. This work was also partly supported in its early stages by the Gordon and Betty Moore Foundation's Data Driven Discovery Initiative through grant GBMF4557 to KAR.

## Author contributions

T.N.N. and K.A.R. conceptualized the work and designed experiments. K.A.R. created the mathematical model. T.N.N. collected all deep mutational scanning data. T.N.N. and K.A.R. analyzed the data. C.I. collected plate-reader based growth rate measurements (used in model

development), performed all Michaelis-Menten enzyme kinetics assays, and performed the TYMS lysate assays. S.M.T. constructed the deep mutational scanning library and contributed to data interpretation. K.A.R. wrote the original draft. T.N.N, C.I., S.M.T. and K.A.R. revised the paper. K.A.R. provided supervision and funding acquisition.

## Competing interests

The authors declare no competing interests.
