## [Peer Review File · Nature Communications]

The Genetic Landscape of a Metabolic InteractionREVIEWER COMMENTS

Reviewer #1 (Remarks to the Author):

This is an excellent manuscript. It describes a nice combination of theory, careful biochemistry, and high-throughput assays to investigate epistasis mediated by a metabolic network. The manuscript was well written, and the work well executed. In general, the modeling and statistics were done convincingly. I have no concerns about the quality or core interpretation of the experimental data. The authors should be congratulated on their excellent and intriguing study.

There are a few areas in the manuscript that I think could be improved. I have organized my comments from (approximately) most to least significant. On the whole, my concerns are minor and mostly have to do with presentation and specific interpretations of the work rather than their core claims.

Major:

1. Of all the modeling presented, their analysis of k_{cat} and enzyme concentration was the least convincing. In some of their analyses, enzyme concentration seemed poorly constrained. It also appeared this value changed depending on what experimental data was used to fit the model. If I understood their model correctly, k_{cat} and [enzyme] are never separated (at least in their in vivo experiments). It might make more sense to just fit V_{max} , which could solve some of their issues with overfitting and with the model changes with increasing amounts of data.

2A. On line 147, the authors write: "While this simplification clearly omits much of folate metabolism, it allows us to write a rate equation that..." The authors should unpack this more. What parts of metabolism are they omitting? How might this simplification cause problems? Why is it justifiable (besides making the math easier)?

2B. Related to 2A, on 117, the authors write that "epistasis is driven by a biochemical interaction, with the added simplification that they are relatively decoupled from the surrounding metabolic context." Do the authors mean that DHFR and TYMS are relatively decoupled, allowing them to isolate the relevant epistasis? Or are the authors asserting that, if they make a simplifying assumption, epistasis exists? This should be clarified. If they mean the latter, they need to argue for the simplifying assumption.

3A. As written, the initial modeling derivation (starting line 119) was a bit confusing. I think this was because the authors led with a description of the variants they used to parameterize the model, jumped back to the model itself, then back to the model derivation. It might help to start with the model derivation, then ask what variants might be useful to parameterize the model, then describe the fitting against those data.

3B. Related to 3A: It took me a bit to figure out what the authors meant when they said that 4 parameters remained (line 163) when it looked like only 3 parameters remained. I had to re-read several

times to realize that the model indeed only had 3 parameters per genotype, but that they were globally fitting 2 of those parameters and then floating 1 parameter in each genotype. A better justification/explanation of this would be helpful. Maybe the authors could explicitly state that the model includes more than one genotype?

4. The authors emphasize that they were able to identify clusters of residues on the structure that correlate with the epistasis. This seems to run counter to their main claim that they have identified epistasis mediated by a metabolic network rather than a direct interaction (which might induce specific patches of structure sensitive to the other genotype). I think it would be much stronger if the authors used their structural data as a *validation* of their other results rather than a new finding. Maybe something like this. Based on our model, we expect residues that muck up enzyme activity to couple strongly in X ways. If we grab the sites that couple in X way in our high-throughput assay, they all cluster around the active site. This provides orthogonal support for our model and high-throughput measurement.

5. In Fig 2D, it looks like there is a “ridge” in epistasis for Q33S/DHFR k_{cat}/K_m and R127A/DHFR k_{cat}/K_m . This seems non-intuitive to me. Why would increasing k_{cat} lower epistasis after a certain point? Maybe the authors commented on this and I missed it, but I think it worth unpacking this further. (Either highlight it as weird and cool or explain why it is trivial and expected).

Minor:

6. Line 26 (and 113, and likely elsewhere): “selected to *emphasize* biochemical epistasis”. The use of emphasize here is awkward. It seems to me the authors are “maximizing” biochemical epistasis here, not emphasizing it. Emphasize implies to me that they are highlighting epistasis out of a sea of non-epistasis; what it seems they are actually doing is selecting conditions where it should have high, experimentally detectable magnitude. [Please take or leave this suggestion]

7. Figure 1B vs. 1D: It took a lot of mental effort to translate between 1B and 1D. Maybe rotate 1B so the DHF and THF and DHFR are in identical places? Also, in the legend, it would help to describe what the points are in 1C, 1E, 1F (“each point corresponds to a genotype...”)

8. Line 230: “negative (or amplifying) epistasis”. This was not clear to me. Is amplifying a synonym for negative epistasis? Or is this contrasting?

Trivial:

9. Line 40: text after semicolon should be a complete sentence. Semicolon should probably be a comma?

10. Line 175: “its’” should be “its”?

Reviewer #2 (Remarks to the Author):

Review for NCOMMS-23-38409

The importance of intermolecular epistasis mediated by intermolecular binding is well established, but mutations in one enzyme can also influence the consequences of mutations in another via the Kacser and Burns framework. More specifically, an enzyme's catalytic flux is determined by its biochemical characteristics such as its k_{cat} and K_m , but it is also influenced by the concentration of its substrate(s) and product(s). Shortages of the former and surpluses of the latter will slow it down, and Kacser and Burns provide an elegant mathematical framing for those effects.

The present study demonstrates epistasis mediated by Kacser-and-Burns style interactions, and complements those high-resolution DMS data with a satisfying quantitative model. The authors ability to map different signs of epistasis to different 3d domains of the enzyme are particularly impressive and satisfying. Overall, I am very positive about this submission, which will be of great interest to diverse investigators, including those interested in protein design, protein biochemistry, and evolutionary genetics among others.

I have only a few comments.

1. Line 42: I suggest deleting the word "both." As I understand the sentence, enzyme velocity is catalytic activity multiplied by enzyme abundance, hence the word "product." But if that reading is correct, the "both" is redundant, and moreover risks confusion, inasmuch as the previous sentence uses the same word for an enzyme's catalytic output.
2. Line 48: "...the relationship between...", but aren't there two relationships here? One between enzymatic velocity and flux, and between enzymatic velocity and cellular growth rate?
3. Lines 55-60: I was surprised to see no mention of a 1993 paper from Eörs Szathmáry in Genetics that makes exactly this point. See <https://doi.org/10.1093/genetics/133.1.127>.
4. Fig 2C: why do model predictions plateau at $\sim 0.8/\text{hr}$ while the data do not?
5. Line 232-233: "Our model accounted for this observation by increasing..." Who's doing the fitting? Is it that the model now predicts a higher concentration, or is this the authors' interpretation?
6. Figure 3c-e: I wonder whether log-transforming the counts might not give us more insight into the component of mutants that substantially disrupt cell growth. And relatedly, mention of "null" mutations on line 281 makes me wish for a column in these histograms that shows me their empirical importance.
7. Figures 5b and e only show epistasis ascribed to TYMS Q33S and R166Q; is that because epistasis is computed in comparison to mutational effects on the TYMS WT background? If so, a mention of this point might be appropriate. Additionally, the authors might be interested in recent quantitative advances that seek to characterize epistasis without requiring the identification of a single benchmark such as this. I wonder whether a so-called "background averaged" methods for computing epistasis might not reveal additional signal in the data. See <https://journals.plos.org/ploscompbiol/article?id=10.1371/journal.pcbi.1004771> for one entrypoint.
8. Fig 5d: Is the horizontal stripe of red dots (line 352) described in Fig 5d related to the source of the plateau mentioned in my point 4 above?
9. Fig 5f/g: perhaps replace 'fraction nativelylike' with "fraction whose growth rate is $\geq 90\%$ of wild type"

or something else that's more explicit?

10. Figure 6a is a wonderfully informative way of representing the situation, but at present, is not at sufficient resolution for detailed examination.

11. Lines 387-388: "consistent with the finding that..." begs the question of mechanism. Even some speculation would be welcome.

12. Figure 7b & d: I would also be interested in seeing the CDF of AA distances to the active site, unconditioned on epistatic category. I suppose it must scale as the volume of a sphere ($4/3\pi r^3$), but of course, AAs aren't quite uniformly packed, and the enzyme isn't a sphere. Anyway, my principled musings are no match for the actual data, which would be nice for comparison to the conditioned CDFs.

13. Line 435: "Extreme loss-of-TYMS function buffered variation in..." Is this use of buffered as in buffering epistasis? I might suggest sticking with the positive/negative language used throughout the rest of the ms.

14. Line 436: "Given these data we expect..." Intriguing but I don't understand the reasoning. I think perhaps the point is that constraint is reduced on this background, but does evolvability mean greater opportunities for adaptation? I am unfortunately not sure why that should be. I'm also not 100% sure about consequences for evolution. If I understand correctly, the point is that the TYMS loss-of-function background increases the number of beneficial mutations. But is it clear that the TYMS loss-of-function mutant won't be dead or nearly so? In other words, I'm thinking that even a profusion of beneficial mutations in this background won't be enough for it to survive in competition with clones carrying a better TYMS allele. In any case, I also counsel using a more explicit word that evolvability, which seems to have dozens of definitions in the literature.

15. Line 437: whatever the disposition of my confusion articulated in the previous point, it is not a "finding" but rather a hypothesis.

16. Line 450-451: What are the prospects for a double-DMS scan, which would pair all single-AA substitutions in DHFR with all single-AA substitutions in TYMS? (Or in DHPS, my personal favorite second partner to DHFR?)

Response to reviewers
“The genetic landscape of a metabolic interaction”

To the reviewers:

Thank you for taking the time to review our manuscript. We were happy to read that you found the work well-executed, the model satisfying, and the results intriguing. We also appreciated your concrete, constructive, and detailed feedback. We revised our manuscript to: 1) include new experimental data testing our model’s predictions, 2) clarify aspects of our modeling approach, and 3) address reviewer comments related to clarity and presentation. Together, we believe these revisions have improved the manuscript and address the reviewer concerns in full. Please find our detailed point-by-point response below. Reviewer comments are in blue with our response in black, we have highlighted modified page numbers and figures in yellow.

Additionally, in line with the Nature Communications formatting recommendations, we have made minor modifications throughout the manuscript to ensure compliance with character limits on subheadings, word limits on figure legends, and the appropriate order of article sections.

Reviewer 1:

Major:

*1. Of all the modeling presented, their analysis of k_{cat} and enzyme concentration was the least convincing. In some of their analyses, enzyme concentration seemed poorly constrained. It also appeared this value changed depending on what experimental data was used to fit the model. If I understood their model correctly, k_{cat} and [enzyme] are never separated (at least in their *in vivo* experiments). It might make more sense to just fit V_{max} , which could solve some of their issues with overfitting and with the model changes with increasing amounts of data.*

Thank you for this comment and the opportunity to further clarify our model. Yes, as you correctly understand, mutational effects on k_{cat} and [enzyme] are not separated by our *in vivo* data. Nonetheless, we believe there is additional information to be had by fitting these parameters separately — especially in cases where we do have experimental *in vitro* constraints on k_{cat} . Moreover, this approach leaves room to improve our model (through the addition of further experimental constraints on abundance) in future work. As we treat the fitting of k_{cat} and [enzyme] slightly differently for DHFR and TYMS, we address our rationale for each of these separately.

Fitting for DHFR: In the case of DHFR, we fit only a single global parameter for [enzyme] that is held constant across all variants. This parameter can be viewed almost as a global scaling factor for the activity of DHFR across the entire experiment. Keeping [enzyme] and k_{cat} separate allows us to: 1) integrate experimentally measured k_{cat} values into the model, and 2) leaves space for us to include experimental measurements of [enzyme] in our future work. For the smaller scale experiments (Fig. 1 d-f – 10 genotypes, Fig. 2c – 28 genotypes) and a subset of the larger deep mutational scanning data set (Fig. 5a – 114 genotypes) we have used experimental *in vitro* measurements of k_{cat} to constrain our model (e.g. k_{cat} is not fit, and only a single [DHFR] parameter is fit). In these cases, fitting only V_{max} as the reviewer suggests would miss available experimental information. The strong agreement between the model and experimental data in Fig. 5a ($R^2 = 0.84$) provides evidence that approximating DHFR intracellular abundance (concentration) with only a single parameter is reasonable, a point we now emphasize in the main text (p.16, lines 400-404). Moreover, deviations between the model

and the data can generate informative hypotheses about mutants that impact stability and/or abundance (rather than catalytic activity).

For DHFR mutants where we lack *in vitro* constraints, we infer the k_{cat} separately. At this point, it is possible that any mutant-dependent effects on abundance become collapsed into our k_{cat} estimate, resulting in an “effective” k_{cat} that is scaled by our global DHFR concentration parameter. While this does introduce a potential source of error, we again see reasonable (if not exact) agreement between experimental and inferred values for k_{cat} (Fig. 5c). We have added additional discussion and clarification of this in the main text (p.16, lines 414-419).

Fitting for TYMS: Unlike the single global parameter for [DHFR], we fit mutant-specific parameters for [TYMS] (see also response to 3B below). Again, we incorporate *in vitro* experimental measurements of TYMS k_{cat} so that only [TYMS] is being fit. Like the reviewers, we noticed that [TYMS] did vary when the model was fit to data from different experiments, but also that: 1) the best-fit values for TYMS mutant concentrations were correlated across model fits, and 2) the fit values consistently made the interesting prediction that TYMS Q33S was expressed at higher abundance than WT (despite being lower activity). Both of these observations convinced us that separately fitting [enzyme] and k_{cat} was informative despite the cross-experiment variation. We now more completely discuss the correlation and variation between [TYMS] values across different model fits (p.15, lines 392-398).

Additionally, we experimentally tested the model prediction that TYMS Q33S is expressed at higher abundance than WT (despite having lower catalytic activity). We were unable to quantify intracellular TYMS by Western blot, as all commercially available TYMS antibodies were developed against human TYMS and did not show sufficient cross-reactivity and sensitivity to detect *E. coli* TYMS in lysates. However, we were able to use activity in crude lysates alongside our *in vitro* k_{cat} data to estimate [TYMS] for the WT, Q33S, R127A, and R166Q variants (following a similar approach as *Rodrigues et al. 2016 PNAS 113:E1470*). These experimental estimates of intracellular abundance were well correlated to our model fits, and showed that Q33S velocity is higher than WT in the conditions of our experiment. These new data are now presented in Fig. 2d-e and discussed on p. 11.

2A. On line 147, the authors write: “While this simplification clearly omits much of folate metabolism, it allows us to write a rate equation that...” The authors should unpack this more. What parts of metabolism are they omitting? How might this simplification cause problems? Why is it justifiable (besides making the math easier)?

Yes – this is a good suggestion, which is especially important for readers unfamiliar with folate metabolism. We now describe the reactions that are being omitted, and more carefully discuss our rationale for doing so on p.6, lines 143-151.

2B. Related to 2A, on 117, the authors write that “epistasis is driven by a biochemical interaction, with the added simplification that they are relatively decoupled from the surrounding metabolic context.” Do the authors mean that DHFR and TYMS are relatively decoupled, allowing them to isolate the relevant epistasis? Or are the authors asserting that, if they make a simplifying assumption, epistasis exists? This should be clarified. If they mean the latter, they need to argue for the simplifying assumption.

We mean that DHFR and TYMS are epistatically coupled to each other, while being relatively decoupled from the remainder of the genes in the pathway. This allows us to isolate epistasis to

the two gene pair (without consideration of couplings to other folate metabolic genes). This claim is supported by prior work from ourselves and others (*Schober et al. 2019 Cell Reports*, *Rodrigues et al. 2019 Elife*) and motivates our construction of a mathematical model that focuses on DHFR/TYMS as an isolated coupled unit. We now clarify this point on p.4 108-112.

3A. As written, the initial modeling derivation (starting line 119) was a bit confusing. I think this was because the authors led with a description of the variants they used to parameterize the model, jumped back to the model itself, then back to the model derivation. It might help to start with the model derivation, then ask what variants might be useful to parameterize the model, then describe the fitting against those data.

We have now re-organized the manuscript to first derive the model (p. 5-6) and then discuss model parameterization (p. 7-8) as suggested. This was a helpful suggestion - we agree that this is more clear!

3B. Related to 3A: It took me a bit to figure out what the authors meant when they said that 4 parameters remained (line 163) when it looked like only 3 parameters remained. I had to re-read several times to realize that the model indeed only had 3 parameters per genotype, but that they were globally fitting 2 of those parameters and then floating 1 parameter in each genotype. A better justification/explanation of this would be helpful. Maybe the authors could explicitly state that the model includes more than one genotype?

Yes, this is an important point. We now spend more time to explain that we fit separate enzyme concentrations for each TYMS variant and only a single global parameter for DHFR concentration. We also describe our rationale, on p. 7-8, lines 183-190. We also touch on this in our response to reviewer 2.

*4. The authors emphasize that they were able to identify clusters of residues on the structure that correlate with the epistasis. This seems to run counter to their main claim that they have identified epistasis mediated by a metabolic network rather than a direct interaction (which might induce specific patches of structure sensitive to the other genotype). I think it would be much stronger if the authors used their structural data as a *validation* of their other results rather than a new finding. Maybe something like this. Based on our model, we expect residues that muck up enzyme activity to couple strongly in X ways. If we grab the sites that couple in X way in our high-throughput assay, they all cluster around the active site. This provides orthogonal support for our model and high-throughput measurement.*

We have now added new text to better emphasize that the structural organization of epistasis around the active site is in line with expectation from our model (p. 18, lines 450-456, 478-480).

5. In Fig 2D, it looks like there is a “ridge” in epistasis for Q33S/DHFR kcat/Km and R127A/DHFR kcat/Km. This seems non-intuitive to me. Why would increasing kcat lower epistasis after a certain point? Maybe the authors commented on this and I missed it, but I think it worth unpacking this further. (Either highlight it as weird and cool or explain why it is trivial and expected).

This is indeed trivial and expected, and very much worth explaining. We provide a more complete explanation of this now on lines 290-294.

Minor:

6. Line 26 (and 113, and likely elsewhere): “selected to **emphasize** biochemical epistasis”. The use of *emphasize* here is awkward. It seems to me the authors are “maximizing” biochemical epistasis here, not *emphasizing* it. *Emphasize* implies to me that they are highlighting epistasis out of a sea of non-epistasis; what it seems they are actually doing is selecting conditions where it should have high, experimentally detectable magnitude. [Please take or leave this suggestion]

We agree the use of *emphasize* is kind of awkward but are somewhat at a loss to find better wording and have kept it. We are indeed selecting conditions where the DHFR/TYMS epistasis should have high, detectable magnitude. However, since we did not do any kind of systematic parameter optimization or search across experimental condition space, I am not at all confident that we maximize epistasis between this gene pair (indeed there are almost certainly more extreme conditions than the ones we choose here).

7. Figure 1B vs. 1D: It took a lot of mental effort to translate between 1B and 1D. Maybe rotate 1B so the DHF and THF and DHFR are in identical places? Also, in the legend, it would help to describe what the points are in 1C, 1E, 1F (“each point corresponds to a genotype...”)

Nice suggestion. We have rotated the schematic in the panel (though now it is panel 1C). We have also added additional description of the data points in the legend for panels d-f (including the wording “Each point is a particular DHFR/TYMS genotype.”)

8. Line 230: “*negative (or amplifying) epistasis*”. This was not clear to me. Is *amplifying* a synonym for *negative epistasis*? Or is this contrasting?

We intended *amplifying* as a synonym, but since this was unclear we have gotten rid of the parenthetical altogether. Now the text simply reads “*negative epistasis*” (line 257).

Trivial:

9. Line 40: text after semicolon should be a complete sentence. Semicolon should probably be a comma?

Yes, we have changed it to a comma (line 38).

10. Line 175: “*its*” should be “*its*”?

Corrected! (line 197)

Reviewer 2:

1. Line 42: I suggest deleting the word “both.” As I understand the sentence, enzyme velocity is catalytic activity multiplied by enzyme abundance, hence the word “product.”

But if that reading is correct, the “both” is redundant, and moreover risks confusion, inasmuch as the previous sentence uses the same word for an enzyme’s catalytic output.

We have deleted “both”, agreed that we want to make clear that by “product” we mean multiplication (line 39).

2. Line 48: “...the relationship between...”, but aren’t there two relationships here? One between enzymatic velocity and flux, and between enzymatic velocity and cellular growth rate?

Yes, there are multiple relationships here, we have modified the text to better indicate plurality (lines 46-47).

3. Lines 55-60: I was surprised to see no mention of a 1993 paper from Eörs Szathmáry in Genetics that makes exactly this point. See <https://doi.org/10.1093/genetics/133.1.127>.

Thank you for suggesting this very appropriate reference. We now cite this work (line 55).

4. Fig 2C: why do model predictions plateau at ~0.8/hr while the data do not?

Yes – good question. We have also noticed this; it seems the best fit parameters for this data set result in a situation where none of the mutants are practically able to reach the maximum growth rate because K (the parameter describing the relative THF concentration at 50% growth) is set fairly high. It seems that this parameter selection is necessary though to resolve growth rate differences amongst the slower growing DHFR/TYMS variant combinations. We now comment on this briefly in the main text (lines 225-227).

5. Line 232-233: “Our model accounted for this observation by increasing...” Who’s doing the fitting? Is it that the model now predicts a higher concentration, or is this the authors’ interpretation?

Thank you for pointing out the need for clarification here. The model predicts a higher concentration; we now state this more explicitly on lines 259-261. During revisions we have added new experiments to test this prediction (see also response to reviewer 1, and manuscript lines 264-282, Fig. 2 d-e)

6. Figure 3c-e: I wonder whether log-transforming the counts might not give us more insight into the component of mutants that substantially disrupt cell growth. And relatedly, mention of “null” mutations on line 281 makes me wish for a column in these histograms that shows me their empirical importance.

These are interesting suggestions. We tried log transforming the counts in panel 3c-e (see rough plots below), but in our opinion these were less clear than the current figure set. The log transformation compressed differences at the high end (large peak) of the data set and made all three data sets appear more similar:

With this in mind, we have left the figure with a linear y-axis. As for the null mutations – the number is so small as to be barely visible on the histograms. However, the raw counts of null mutations in each TYMS background is consistent with the overall pattern of epistasis. There are (on average) 41 null mutations per experimental replicate in the WT TYMS context, 82 in the TYMS Q33S context (negative epistasis), and 7 in the R166Q TYMS context (positive or buffering epistasis). We have added these numbers to the main text of our manuscript (lines 343-345).

7. Figures 5b and e only show epistasis ascribed to TYMS Q33S and R166Q; is that because epistasis is computed in comparison to mutational effects on the TYMS WT background? If so, a mention of this point might be appropriate. Additionally, the authors might be interested in recent quantitative advances that seek to characterize epistasis

without requiring the identification of a single benchmark such as this. I wonder whether a so-called “background averaged” methods for computing epistasis might not reveal additional signal in the data.

See <https://journals.plos.org/ploscompbiol/article?id=10.1371/journal.pcbi.1004771> for one entrypoint.

Yes, we do compute epistasis relative to the WT TYMS background. This is an important clarification, and a point we now explicitly make on lines 349 and 398. We are familiar with “background averaged” methods and find the ideas in Poelwijk et al. very interesting. We may consider this in future work, particularly in situations where we have data in more than only three genetic backgrounds.

8. Fig 5d: Is the horizontal stripe of red dots (line 352) described in Fig 5d related to the source of the plateau mentioned in my point 4 above?

Interesting thought, and it is hard to say for sure given current data. The stripe of red dots you describe are predicted to grow like wildtype, yet they actually grow more slowly. Our favored explanation is that these variants have a deleterious effect on growth that is not accounted for by our model, which is the explanation we put forth in the manuscript (lines 425-427). However, some of this plateau effect may come from the steep relationship between THF abundance and growth rate (as shown in Fig. 1d) and sensitivity of the model to changes in K (the location of the steep part of the THF/growth rate curve).

9. Fig 5f/g: perhaps replace “fraction nativelylike” with “fraction whose growth rate is $\geq 90\%$ of wild type” or something else that’s more explicit?

We found it challenging to fit this additional text into the figure, but note that the legend does state “The heatmap shows the fraction of DHFR mutations with growth rates of 0.9 or better”.

10. Figure 6a is a wonderfully informative way of representing the situation, but at present, is not at sufficient resolution for detailed examination.

We agree that the labels in Fig. 6a are extremely small. Our hope is PDF versions of the article will allow interested readers to zoom in and view these labels more carefully. We have increased the resolution of the embedded tiff in our main manuscript file, and will supply high resolution vector graphics with our revised manuscript.

11. Lines 387-388: “consistent with the finding that...” begs the question of mechanism. Even some speculation would be welcome.

Thank you for the invitation to speculate! We have now added two sentences of text in this direction on lines 466-469.

12. Figure 7b & d: I would also be interested in seeing the CDF of AA distances to the active site, unconditioned on epistatic category. I suppose it must scale as the volume of a sphere ($4/3\pi r^3$), but of course, AAs aren’t quite uniformly packed, and the enzyme isn’t a sphere. Anyway, my principled musings are no match for the actual data, which would be nice for comparison to the conditioned CDFs.

We have added curves describing the CDF of all amino acids distances to the active site to **Fig. 7 b and d**. (Please see the new black dashed lines, these are indeed nice for comparison).

13. Line 435: “Extreme loss-of-TYMS function buffered variation in...” Is this use of buffered as in buffering epistasis? I might suggest sticking with the positive/negative language used throughout the rest of the ms.

Yes, we meant this as in buffering epistasis. To improve clarity, we have modified this to read “Extreme loss-of-TYMS function **rescued strongly deleterious mutations...**” (line 523).

14. Line 436: “Given these data we expect...” Intriguing but I don’t understand the reasoning. I think perhaps the point is that constraint is reduced on this background, but does evolvability mean greater opportunities for adaptation? I am unfortunately not sure why that should be. I’m also not 100% sure about consequences for evolution. If I understand correctly, the point is that the TYMS loss-of-function background increases the number of beneficial mutations. But is it clear that the TYMS loss-of-function mutant won’t be dead or nearly so? In other words, I’m thinking that even a profusion of beneficial mutations in this background won’t be enough for it to survive in competition with clones carrying a better TYMS allele. In any case, I also counsel using a more explicit word that evolvability, which seems to have dozens of definitions in the literature.

Yes, the point is that the TYMS loss-of-function background reduces constraints on the DHFR sequence and expands the space of possible mutations that could now be used to vary functional properties like drug resistance or substrate specificity. Interestingly, TYMS loss-of-function mutations can be viable evolutionary solutions depending on environmental context: trimethoprim resistant clinical isolates of *E. coli* are sometimes thymidine auxotrophs (*King et al. 1983, J. Clinical Microbiology 18:79*). This indicates that in some situations a TYMS null allele is preferred to a variant with native TYMS. We have added this information to **lines 530-532** of the main text. Regardless of the natural evolution case, we imagine using TYMS as a knob to tune selective pressure on DHFR in laboratory screens and/or evolution – a situation that could be useful for experimentally testing computationally designed synthetic DHFR sequences.

15. Line 437: whatever the disposition of my confusion articulated in the previous point, it is not a “finding” but rather a hypothesis.

Yes, this is definitely true, we have modified the wording on **lines 526 and 528** to make this more clear.

16. Line 450-451: What are the prospects for a double-DMS scan, which would pair all single-AA substitutions in DHFR with all single-AA substitutions in TYMS? (Or in DHPS, my personal favorite second partner to DHFR?)

That would be quite interesting. It is not experimentally feasible (at present) to explore the complete space of double mutations given the intersection of combinatorial complexity and next generation sequencing costs. A complete double DMS library spanning all pairwise combinations of 3,180 DHFR single mutations with 5,280 TYMS single mutations would come to just under 17 million combinations – well above the typical DMS screen size of $\sim 10^3$ - 10^4

variants. However, one could get quite far by having a more specific experimental design that focused on more evolutionarily conserved positions, or did not consider all possible mutations per site. We now comment on this in the discussion (lines 512-518).

REVIEWERS' COMMENTS

Reviewer #1 (Remarks to the Author):

The authors have adequately addressed my concerns from the initial submission. I'd like to congratulate them on an excellent piece of work.

Reviewer #1 (Remarks on code availability):

I was able to run most of the code and reproduce most of their analyses. (Very straightforward notebooks!)

Only problem I had was that the code looked for 'Output/grates.pkl'. This was not in the repo, meaning I could not run most of 3_Epistasis.ipynb.

Reviewer #2 (Remarks to the Author):

I thank the authors for their thoughtful responses to my earlier concerns and suggestions. These are now fully resolved, and I have no further reservations about this exciting contribution.

Response to reviewers
“The genetic landscape of a metabolic interaction”

To the Reviewers:

Thank you for taking the time to review our revised manuscript and evaluate our associated code on github. We have addressed the last remaining reviewer concern, please see below for our brief response.

Reviewer #1 (Remarks on code availability):

I was able to run most of the code and reproduce most of their analyses. (Very straightforward notebooks!)

Only problem I had was that the code looked for 'Output/grates.pkl'. This was not in the repo, meaning I could not run most of 3_Epistasis.ipynb.

We are glad you found the notebooks straightforward. We have added the files Output/grates.pkl and Outputs/Epistasis.pkl to our github repository. In addition to these compressed pickle (*.pkl) files, we also include the growth rate and epistasis results as tab-delimited text. We additionally note that the pickle and tab delimited files should also be auto-generated by the code when each notebook (labeled 1-4) is run sequentially and completely.

Thank you again for a productive review process,

Kim Reynolds